# Computing Exact Shapley Values in Polynomial Time for Product-Kernel Methods

## Abstract

Kernel methods are widely used in machine learning due to their flexibility and expressiveness. However, their black-box nature poses significant challenges to interpretability, limiting their adoption in high-stakes applications. Shapley value-based feature attribution techniques, such as SHAP and kernel method-specific adaptation like RKHS-SHAP, offer a promising path toward explainability. Yet, computing exact Shapley values is generally intractable, leading existing methods to rely on approximations and thereby incur unavoidable error. In this work, we introduce PKeX-Shapley, a novel algorithm that utilizes the multiplicative structure of product kernels to enable the exact computation of Shapley values in polynomial time. The core of our approach is a new value function, the *functional baseline value function*, specifically designed for product-kernel models. This value function removes the influence of a feature subset by setting its functional component to the least informative state. Crucially, it allows a recursive thus efficient computation of Shapley values in polynomial time. As an important additional contribution, we show that our framework extends beyond predictive modeling to statistical inference. In particular, it generalizes to popular kernel-based discrepancy measures such as the Maximum Mean Discrepancy (MMD) and the Hilbert–Schmidt Independence Criterion (HSIC), thereby providing new tools for interpretable statistical inference.

## 1 Introduction

Shapley values (Shapley, 1953), a solution concept originating from cooperative game theory, offer a principled, axiomatic framework for feature attribution in machine learning (ML) (Lundberg and Lee, 2017; Covert et al., 2020). Thanks to their rigorous axiomatic foundation, there has been a widespread adoption within the explainable AI community (Sundararajan and Najmi, 2020). They allow a model's output—such as predictions or losses—to be *fairly* distributed across input features based on their individual and joint contributions. Consequently, several algorithms have been proposed to estimate Shapley values under different modelling scenarios, ranging from model-agnostic methods like Kernel SHAP (Lundberg and Lee, 2017) to model-specific methods that leverage structural properties to improve statistical or computational efficiency. Well-known examples of the latter include Tree SHAP for tree-based models (Lundberg et al., 2020), GPSHAP for Gaussian processes (Chau et al., 2024), Deep SHAP for deep networks (Lundberg and Lee, 2017), and, most relevant to our work, RKHS-SHAP for kernel methods (Chau et al., 2022). Kernel methods are particularly notable—not only for their use in prediction tasks but also in a broad range of statistical inference problems, including measuring distributional closeness (Gretton et al., 2006; Naslidnyk et al., 2025), two-sample testing (Schrab, 2025; Chau et al., 2025), goodness-of-fit testing (Chwialkowski et al., 2016; Liu et al., 2016), independence testing (Albert et al., 2022), causal inference and discovery (Mitrovic et al., 2018; Chau et al., 2021; Sejdinovic, 2024), among others. Hence, as kernel methods gain widespread adoption in high-stakes applications for their flexibility and expressive power, the need for interpretability has become increasingly vital.

Despite their game-theoretic foundation, Shapley values face two major challenges when applied to machine learning. The first is defining and estimating a suitable value function that quantifies the contribution of a coalition of features. Ideally, this function should capture the model's behavior when the complementary features are absent. A natural approach is to retrain the model on each subset and use the resulting prediction as the value (Lipovetsky and Conklin, 2001), but this is

computationally infeasible due to the exponential number of retrainings required. A more common alternative simulates feature absence through marginal or conditional expectations of the model's output, given the fixed subset (see Sundararajan and Najmi (2020) for other formulations). However, estimating these expectations reliably is difficult, as it often involves density estimation or simplifying assumptions such as feature independence—assumptions that can lead to misleading and unfaithful explanations (Kumar et al., 2020). Chau et al. (2022) tackle this problem in the context of kernel methods by exploiting the structure of reproducing kernel Hilbert spaces (RKHS) and using kernel distributional embeddings (Muandet et al., 2017) to estimate value functions nonparametrically, thereby avoiding density estimation and independence assumptions.

Once a value function is fixed, the second challenge lies in efficiently estimating the Shapley values themselves. Exact computation requires evaluating the value function over all $2^d$ possible feature subsets for $d$ features, which is computationally demanding. To alleviate this, approximation techniques such as Monte Carlo sampling and regression-based methods are widely used, relying on a smaller set of evaluations. While these approaches reduce computational costs, they inevitably introduce estimation errors that grow with both sample size and feature dimensionality (Kumar et al., 2020). In some cases, model-specific structures can be exploited for efficiency—for example, Tree SHAP leverages tree decompositions for exact polynomial-time computation. By contrast, RKHS-SHAP improves the statistical estimation of the value function but still relies on regression-based approximations for the Shapley values themselves, and thus inherits the same computational limitations.

To design an efficient Shapley value–based attribution algorithm for kernel methods, we focus on a subclass that employs product kernels—referred to as *product-kernel methods*—and introduce *PKeX-Shapley* (**P**roduct-**K**ernel-based **eX**act **Shapley** attribution). Product kernels are particularly attractive as they are simple to implement while retaining strong theoretical guarantees. For instance, the product of universal kernels remains universal, so the resulting RKHS can approximate any continuous function provided that each component kernel is sufficiently expressive (Szabó and Sriperumbudur, 2018). Although this focus narrows the scope compared with Chau et al. (2022), which considers generic kernels, the additional structure enables us to directly address the two challenges outlined

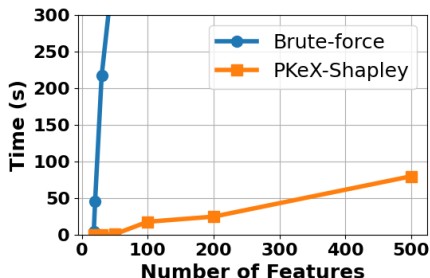

Figure 1: Execution time of brute-force and our PKeX-Shapley with a 300s budget (see details in Appendix H.2).

above. Specifically, we introduce a new value function, called the *functional baseline value function*, which leverages the multiplicative structure of product kernels. Conceptually, it parallels baseline value functions: rather than setting features to fixed baseline inputs, we set the corresponding kernel components to their least informative state, thereby eliminating their influence. This design admits a natural interpretation as the optimal orthogonal projection of the predictive function onto a function class restricted to a given feature subset. Crucially, this formulation enables a recursive algorithm for computing exact Shapley values in polynomial time (see Figure 1 for a runtime comparison with naïve computation). This yields substantial efficiency gains without relying on approximation techniques such as sampling or regression-based estimation.

As an additional, independent contribution, we demonstrate that our framework extends beyond kernel-based predictive models. In particular, we show how our method applies to kernel-based statistical discrepancies, including the Maximum Mean Discrepancy (MMD) and Hilbert–Schmidt Independence Criterion (HSIC), thereby enabling interpretable statistical inference. A Python implementation of our method is publicly available (pke, 2025).

## 2 BACKGROUND ON SHAPLEY VALUES

**Notation.** Let $\mathcal{D}$ denote the set of $d$ features, and $2^{\mathcal{D}}$ its power set. The training set $\{(\boldsymbol{x}^{(i)}, y^{(i)})\}_{i=1}^{n}$ consists of $n$ samples, where $\boldsymbol{x}^{(i)} \in \mathbb{R}^d$ and $y^{(i)} \in \mathbb{R}$ (or a discrete label set for classification tasks). Let $\mathbf{X} \in \mathbb{R}^{n \times d}$ be the matrix of features, and $\mathbf{X}_{\mathcal{S}} \in \mathbb{R}^{n \times s}$ the submatrix restricted to features in subset $\mathcal{S} \subseteq \mathcal{D}$, and we write $\mathbf{X}_j := \mathbf{X}_{\{j\}}$. We use capital letters for random variables, bold capital letters for matrices, calligraphic letters for sets, and bold lowercase letters for vectors. The restriction

of a vector $\boldsymbol{x}$ to features in $\mathcal{S}$ is denoted by $\boldsymbol{x}_{\mathcal{S}}$. The elementwise product is denoted by $\odot$, and expectation by $\mathbb{E}$. A symmetric positive (semi-)definite kernel function over $\mathcal{D}$ is denoted by $k$, and its restriction to subset $\mathcal{S}$ by $k_{\mathcal{S}}$. The corresponding kernel matrices are denoted by $\mathbf{K}$ and $\mathbf{K}_{\mathcal{S}}$, respectively. All proofs are provided in Appendix D.

**Shapley value.** The Shapley value (Shapley, 1953) is a widely used game-theoretic method for feature attribution in predictive models. It assigns importance to each input feature by averaging its marginal contribution across all possible feature subsets, with weights determined by coalition size. This construction uniquely satisfies several fairness axioms, making it a principled approach to distributing a model's output among its features. Specifically, given a value function $v : 2^{\mathcal{D}} \to \mathbb{R}$, which quantifies the contribution of a feature subset, the Shapley value for a feature $j$ is defined as

$$\phi_j = \sum_{\mathcal{S} \subseteq \mathcal{D} \setminus \{j\}} \mu(|\mathcal{S}|)\big(v(\mathcal{S} \cup \{j\}) - v(\mathcal{S})\big), \tag{1}$$

where $\mu(s) = {}^{s!(d-s-1)!}/{}_{d!}$ and $v(\mathcal{S} \cup \{j\}) - v(\mathcal{S})$ measures the marginal contribution of feature $j$ when added to subset $\mathcal{S}$. The Shapley value is the unique solution satisfying four axioms of efficiency, null player, symmetry, and linearity for any given value function (Shapley, 1953). However, in the context of explainability, defining this value function $v$—which determines how subsets of features contribute to the model's output—is a non-trivial design choice. Several formulations have been proposed (Sundararajan and Najmi, 2020), but two are widely adopted: the interventional value function $v_{\boldsymbol{x}}(\mathcal{S}) = \mathbb{E}_{X_{\mathcal{D} \setminus \mathcal{S}} | X_{\mathcal{S}} = \boldsymbol{x}_{\mathcal{S}}}[f(\boldsymbol{x}_{\mathcal{S}}, X_{\mathcal{D} \setminus \mathcal{S}})]$, which replaces missing features with samples from their marginal distributions (Janzing et al., 2020), and the observational value function $v_{\boldsymbol{x}}(\mathcal{S}) = \mathbb{E}_{X_{\mathcal{D} \setminus \mathcal{S}}}[f(X) \,|\, X_{\mathcal{S}} = \boldsymbol{x}_{\mathcal{S}}]$, which imputes them using the conditional distribution given observed features (Lundberg and Lee, 2017). While the observational approach preserves feature dependencies, it requires estimating complex conditional expectations, making it computationally demanding. In practice, both approaches are often implemented by sampling from marginal or conditional distributions, requiring repeated model evaluations that become costly for large models or high-dimensional inputs (Sundararajan and Najmi, 2020). Moreover, these estimates typically rely on a finite set of background samples (commonly drawn from the training set), which can substantially influence the resulting explanations (Molnar, 2023, Chapter 21).

## 3 Exact Shapley value for product-kernel learning methods

In this section, we demonstrate how to compute *exact* Shapley values for local explanation in polynomial time when the predictive model is constructed through product-kernel methods, e.g., an SVM or kernel ridge regressor built using product kernels.

### 3.1 A new functional baseline value function for product-kernel methods

Product-kernel methods rely on kernel functions to capture complex relationships between input features and output. A kernel-based decision function is generally expressed as

$$f(\boldsymbol{x}) = \sum_{i=1}^{n} \alpha_i k(\boldsymbol{x}, \boldsymbol{x}^{(i)}) = \boldsymbol{\alpha}^{\top} k(\mathbf{X}, \boldsymbol{x})$$

where $k$ is a product kernel function, and $\alpha_i$ are the learned coefficients associated with the model. Recall that the product kernel $k$ can be expressed as products of base kernels $k_j, j \in \mathcal{D}$:

$$k(\boldsymbol{x}, \boldsymbol{x}^{(i)}) = \prod_{j \in \mathcal{D}} k_j(x_j, x_j^{(i)}). \tag{2}$$

Product-kernel methods are widely used in machine learning for their simplicity and effectiveness in modeling similarities in high-dimensional data—by designing a kernel for each feature and then multiplying them together (Gardner et al., 2018). They also come with strong theoretical guarantees: if the base kernels are universal—i.e., capable of approximating any continuous function defined on the marginal input—then the product kernel inherits this property (Szabó and Sriperumbudur, 2018). Some well-known kernels, such as the radial basis function (RBF) with an isotropic or anisotropic bandwidth, belong to the family of product kernels, i.e., $k(\boldsymbol{x}, \boldsymbol{x}') = \exp(-\|\boldsymbol{x} - \boldsymbol{x}'\|^2/2\sigma^2) = \prod_{j=1}^{d} \exp(-(x_j - x_j')^2/2\sigma^2)$ (see Appendix A for more details on product kernels).

We now introduce our *functional baseline value function*, motivated by the multiplicative structure of product kernels. In standard settings, interventional value functions remove the effect of a feature by replacing it with its average contribution, thereby isolating its marginal impact. More generally, this can be viewed as a baseline value function (Sundararajan and Najmi, 2020, Sec. 2.3), where features are set to a non-informative state, such as zero vectors in image data. In our setting, however, the functional form of the predictive model is directly accessible. We therefore remove a feature's influence by factoring out its corresponding functional component and setting it to one—the least informative contribution in the multiplicative structure. This design principle yields the following definition for product-kernel methods.

**Definition 1.** *Let $\boldsymbol{\alpha}$ be the coefficients of the function $f$ constructed using the product-kernel method. The functional baseline value function for an input $\boldsymbol{x}$, a feature coalition $\mathcal{S}$, and coefficients $\boldsymbol{\alpha}$ is*

$$\nu_{\boldsymbol{x}}(\mathcal{S}) = \boldsymbol{\alpha}^\top k_{\mathcal{S}}(\mathbf{X}_{\mathcal{S}}, \boldsymbol{x}_{\mathcal{S}}). \tag{3}$$

Notably, computing this value function requires only linear time, making it highly efficient. A detailed comparison with baseline and interventional value functions is provided in Appendix B.

We emphasize that the choice of a value function determines how feature subset importance is measured, and is therefore inherently subjective. Our introduction of the functional baseline value function is motivated not only by its conceptual alignment with existing formulations, but also by the substantial computational benefits it enables, as we will demonstrate in the following subsection. We do not claim that this value function is intrinsically superior in capturing feature importance; however, we demonstrate in Experiment #2 (Section 5.1) that its empirical performance in recovering influential features is promising and on par with other established baselines.

## 3.2 A RECURSIVE FORMULATION FOR COMPUTING SHAPLEY VALUES

Having motivated our functional baseline value function, we now demonstrate the computational gain we obtain from this formulation.

**Theorem 2.** *Let $\mathcal{Z} := \{\boldsymbol{z}_1, \ldots, \boldsymbol{z}_d\}$ with $\boldsymbol{z}_i = k_i(\mathbf{X}_i, x_i)$ and $e_q(\mathcal{Z})$ the elementary symmetric polynomials (ESPs) of order $q$ over $\mathcal{Z}$, defined recursively as*

$$e_q(\mathcal{Z}_{-j}) = \frac{1}{q} \sum_{r=1}^{q} (-1)^{r-1} e_{q-r}(\mathcal{Z}_{-j}) \odot p_r(\mathcal{Z}_{-j}),$$

*where $p_r(\mathcal{Z}) = \sum_{\boldsymbol{z}_i \in \mathcal{Z}} \boldsymbol{z}_i^r$ is the element-wise degree-$r$ power sum. For product-kernel learning methods with coefficients $\boldsymbol{\alpha}$, the Shapley value $\phi_j^{\boldsymbol{x}}$ for feature $j$ of instance $\boldsymbol{x}$ can then be expressed as*

$$\phi_j^{\boldsymbol{x}} := \sum_{\mathcal{S} \subseteq \mathcal{D} \setminus \{j\}} \mu(|\mathcal{S}|)\big(v_{\boldsymbol{x}}(\mathcal{S} \cup \{j\}) - v_{\boldsymbol{x}}(\mathcal{S})\big) = \boldsymbol{\alpha}^\top \left( \big(k_j(\mathbf{X}_j, x_j) - \mathbf{1}\big) \bigodot \sum_{q=0}^{d-1} \mu(q) e_q\big(\mathcal{Z}_{-j}\big) \right). \tag{4}$$

The recursion follows from Newton's identities for ESPs (Egge, 2019), which due to space constraints, is explained further in Appendix C. Here we provide the key intuition behind equation equation 4. The multiplicative structure of the product kernel allows us to factorize the marginal contribution $v_{\boldsymbol{x}}(\mathcal{S} \cup \{j\}) - v_{\boldsymbol{x}}(\mathcal{S})$ as $\boldsymbol{\alpha}^\top((k_j(\mathbf{X}_j, x_j) - 1) \odot k_{\mathcal{S}}(\mathbf{X}_{\mathcal{S}}, \boldsymbol{x}_{\mathcal{S}}))$. This structure allows us then to push the summation inside the inner product between the coefficients $\boldsymbol{\alpha}$ and the kernel evaluations, and express the total sum over all subsets $\mathcal{S} \subseteq \mathcal{D} \setminus \{j\}$ in terms of weighted ESPs $e_q(\mathcal{Z}_{-j})$, which can then be computed recursively in $O(d^2)$. A similar trick has been used in the context of additive Gaussian processes (Duvenaud et al., 2011), but to our knowledge, it has not been previously leveraged for computing Shapley values in polynomial time. The full algorithm is given in Appendix E, together with a numerically stable modification.

Next, we show the additivity of the explanation for the value function in equation 3 (see Appendix F for discussion on the additivity of explanations for different values for the null game).

**Lemma 3.** *For any instance $\boldsymbol{x}$, the sum of Shapley values satisfies: $\sum_{j=1}^{d} \phi_j^{\boldsymbol{x}} = f(\boldsymbol{x}) - f_{\emptyset}(\boldsymbol{x})$ where $f_{\emptyset}(\boldsymbol{x}) = \sum_{i=1}^{n} \alpha_i$ represents the baseline contribution with no features.*

## 4 Explaining kernel-based statistical discrepancies

While most explainability research has focused on predictive models[1], far less attention has been given to statistical inference tasks, where kernel methods are equally powerful. To both demonstrate the flexibility of our framework and advocate for interpretable statistical inference, we show how it applies to the Maximum Mean Discrepancy (MMD) (Gretton et al., 2006) and the Hilbert–Schmidt Independence Criterion (HSIC) (Gretton et al., 2007), two widely used measures of distributional discrepancy and independence.

### 4.1 Distributing the discrepancy: Explaining the MMD

The *Maximum Mean Discrepancy* (MMD) quantifies the difference between two probability distributions in terms of their kernel mean embeddings: $\text{MMD}^2(\mathbb{P}, \mathbb{Q}) := \|\mu_\mathbb{P} - \mu_\mathbb{Q}\|^2_{\mathcal{H}_k}$, where $\mathcal{H}_k$ is the RKHS associated with the kernel $k$ and $\mu_\mathbb{P} := \int k(\boldsymbol{x}, \cdot) \, d\mathbb{P}(\boldsymbol{x}) \in \mathcal{H}_k$ the kernel mean embedding of $\mathbb{P}$ (Muandet et al., 2017). The embedding $\mu_\mathbb{Q}$ is defined analogously. Based on the samples $\{\boldsymbol{x}^{(i)}\}_{i=1}^n \overset{i.i.d}{\sim} \mathbb{P}$ and $\{\boldsymbol{z}^{(i)}\}_{i=1}^m \overset{i.i.d}{\sim} \mathbb{Q}$, the empirical estimate of MMD, expressed entirely in terms of $k$, is given by $\widehat{\text{MMD}}^2(\mathbb{P}, \mathbb{Q}) = \frac{1}{n(n-1)} \sum_{i \neq j} k(\boldsymbol{x}^{(i)}, \boldsymbol{x}^{(j)}) + \frac{1}{m(m-1)} \sum_{i \neq j} k(\boldsymbol{z}^{(i)}, \boldsymbol{z}^{(j)}) - \frac{2}{nm} \sum_{i,j} k(\boldsymbol{x}^{(i)}, \boldsymbol{z}^{(j)})$.

In this section, we propose an attribution method to allocate the overall discrepancy measured by MMD among the involved variables. Such an attribution is useful in many problems, including explaining MMD-based statistics for hypothesis testing (e.g., Gretton et al. (2012)) and determining the contribution of variables to covariance shift (e.g., Zhang et al. (2020)). Similar to the motivation presented in Section 3, as inspired by the multiplicative structure of product kernels, we define the value function of a coalition $\mathcal{S} \subseteq \mathcal{D}$ as the MMD computed only on the kernel components corresponding to $\mathcal{S}$, with the remaining components set to their least informative state (equal to 1 since we are doing multiplication). This isolates the discrepancy attributable to $\mathcal{S}$ alone.

**Definition 4.** *For a coalition $\mathcal{S} \subseteq \mathcal{D}$, the functional baseline value function for MMD is defined as*
$$v_{MMD}(\mathcal{S}) = \frac{1}{n(n-1)} \sum_{i \neq j} k_\mathcal{S}(\boldsymbol{x}_\mathcal{S}^{(i)}, \boldsymbol{x}_\mathcal{S}^{(j)}) + \frac{1}{m(m-1)} \sum_{i \neq j} k_\mathcal{S}(\boldsymbol{z}_\mathcal{S}^{(i)}, \boldsymbol{z}_\mathcal{S}^{(j)}) - \frac{2}{nm} \sum_{i,j} k_\mathcal{S}(\boldsymbol{x}_\mathcal{S}^{(i)}, \boldsymbol{z}_\mathcal{S}^{(j)}).$$

By applying the same trick as in Theorem 2 to the value function $v_{\text{MMD}}$, we can obtain a similar recursive formulation of Shapley values for MMD. That is, for the first term in $v_{\text{MMD}}$, we use the multiplicate structure of product kernels and write $v_{\text{MMD}}(\mathcal{S} \cup \{q\}) - v_{\text{MMD}}(\mathcal{S})$ as $k_{\mathcal{S} \cup \{q\}}(\boldsymbol{x}_{\mathcal{S} \cup \{q\}}^{(i)}, \boldsymbol{x}_{\mathcal{S} \cup \{q\}}^{(j)}) - k_\mathcal{S}(\boldsymbol{x}_\mathcal{S}^{(i)}, \boldsymbol{x}_\mathcal{S}^{(j)}) = (k_q(x_q^{(i)}, x_q^{(j)}) - 1) k_\mathcal{S}(\boldsymbol{x}_\mathcal{S}^{(i)}, \boldsymbol{x}_\mathcal{S}^{(j)})$. By pushing out $k_q(x_q^{(i)}, x_q^{(j)}) - 1$ from the summation in the Shapley value, we can express the total sum over $\mathcal{S} \subseteq \mathcal{D} \setminus \{q\}$ as the weighted ESPs. The following proposition summarizes this.

**Proposition 5.** *Let $\mathcal{Z}^{(\boldsymbol{x}, \boldsymbol{x}')} = \{k_1(x_1, x_1'), \ldots, k_d(x_d, x_d')\}$, and $e_r(\mathcal{Z}^{(\boldsymbol{x}, \boldsymbol{x}')})$ determined as*
$$e_r(\mathcal{Z}_{-q}^{(\boldsymbol{x}, \boldsymbol{x}')}) = \frac{1}{r} \sum_{s=1}^r (-1)^{s-1} e_{r-s}(\mathcal{Z}_{-q}^{(\boldsymbol{x}, \boldsymbol{x}')}) p_s(\mathcal{Z}_{-q}^{(\boldsymbol{x}, \boldsymbol{x}')}),$$
*where $p_s(\mathcal{Z}) = \sum_{z \in \mathcal{Z}} z^s$ represents the degree-$s$ power sum. Further, let $\gamma_q(\boldsymbol{x}, \boldsymbol{x}')$ be defined as $\gamma_q(\boldsymbol{x}, \boldsymbol{x}') := (k_q(x_q, x_q') - 1) \sum_{r=0}^{d-1} \mu(r) e_r(\mathcal{Z}_{-q}^{(\boldsymbol{x}, \boldsymbol{x}')})$. Then, for product kernels, the Shapley value for the MMD can be recursively computed as*
$$\phi_q^{MMD} = \frac{1}{n(n-1)} \sum_{i \neq j} \gamma_q(\boldsymbol{x}^{(i)}, \boldsymbol{x}^{(j)}) + \frac{1}{m(m-1)} \sum_{i \neq j} \gamma_q(\boldsymbol{z}^{(i)}, \boldsymbol{z}^{(j)}) - \frac{2}{nm} \sum_{i,j} \gamma_q(\boldsymbol{x}^{(i)}, \boldsymbol{z}^{(j)}).$$

The Shapley values $\phi_q^{\text{MMD}}$ allow us to allocate the overall distributional discrepancy between $\mathbb{P}$ and $\mathbb{Q}$ across the variables, identifying the most influential ones in distinguishing the two distributions.

### 4.2 Distributing the dependence: Explaining the HSIC

The *Hilbert-Schmidt Independence Criterion* (HSIC) is a kernel-based dependence measure that quantifies the statistical dependence between two random variables. Let $X$ and $Y$ be two random variables with $k(\cdot, \cdot)$ and $l(\cdot, \cdot)$ as reproducing (product) kernels defined on them. Then, $\text{HSIC}(X, Y) :=$

---

[1]Fleissner et al. (2024) designed an explanation algorithm for kernel-based unsupervised learning.

$\|\mathcal{C}_{XY}\|_{\text{HS}}^2$, where $\mathcal{C}_{XY}$ is the cross-covariance operator and $\|\cdot\|_{\text{HS}}$ is the Hilbert-Schmidt (HS) norm; see, e.g., (Muandet et al., 2017, Sec. 3.6) for technical details. Given a sample $\{(\boldsymbol{x}^{(i)}, \boldsymbol{y}^{(i)})\}_{i=1}^n \overset{i.i.d}{\sim} \mathbb{P}(X, Y)$, $\text{HSIC}(X, Y)$ can be estimated as $\widehat{\text{HSIC}}(X, Y) = (n-1)^{-2}\text{tr}(\mathbf{K}\mathbf{H}\mathbf{L}\mathbf{H})$ where $\mathbf{K} \in \mathbb{R}^{n \times n}$ is the kernel matrix computed using the kernel $k$, i.e., $\mathbf{K}_{ij} = k(\boldsymbol{x}^{(i)}, \boldsymbol{x}^{(j)})$, $\mathbf{L} \in \mathbb{R}^{n \times n}$ is the kernel matrix computed using the kernel $l$, i.e, $\mathbf{L}_{ij} = l(\boldsymbol{y}^{(i)}, \boldsymbol{y}^{(j)})$, and $\mathbf{H} = I - \frac{1}{n}\mathbf{1}\mathbf{1}^\top$ is the centering matrix ensuring zero mean in the feature space.

Computing $\text{HSIC}(X, Y)$ gives us the overall dependence between $X$ and $Y$. We now focus on scenarios where $X$ represents a random variable of the feature vector $\boldsymbol{x}$ and $Y$ represents the scalar prediction outcome $y$. Our interest is then to distribute the overall dependence over the features in $\boldsymbol{x}$. This is particularly useful for feature selection and global sensitivity analysis, where the target is usually univariate. Analogous to the predictive function and MMD cases, we define the functional baseline value function of a coalition $\mathcal{S} \subseteq \mathcal{D}$ as the HSIC computed only with the kernel restricted to $\mathcal{S}$, with all other kernel components set to one. This isolates the dependence captured by $\mathcal{S}$ alone:

**Definition 6.** *For a coalition $\mathcal{S} \subseteq \mathcal{D}$, the functional baseline value function for HSIC is defined as $v_{HSIC}(\mathcal{S}) = \frac{1}{(n-1)^2}\text{tr}(\mathbf{H}\mathbf{L}\mathbf{H}\,\mathbf{K}_\mathcal{S})$, where $\mathbf{K}_\mathcal{S} = \bigodot_{j \in \mathcal{S}} \mathbf{K}_j$ is the Hadamard product of kernel matrices restricted to $\mathcal{S}$.*

This construction follows the same principle as before: setting kernel contributions of features outside $\mathcal{S}$ to their least informative value (equal to 1 as we are doing multiplication). Thus $v_{\text{HSIC}}(\mathcal{S})$ quantifies the dependency between $X_\mathcal{S}$ and $Y$, independently of the remaining features. Applying the same trick as in Proposition 5 to the value function $v_{\text{HSIC}}$ yields a similar recursive formulation of Shapley values for HSIC.

**Proposition 7.** *For the product kernel, the Shapley value for HSIC can be recursively computed as:*

$$\phi_j^{HSIC} = \frac{1}{(n-1)^2}\text{tr}\left(\mathbf{H}\mathbf{L}\mathbf{H}\big((\mathbf{K}_j - \mathbf{1}\mathbf{1}^\top)\bigodot\sum_{q=0}^{d-1}\mu(q)E_q(\mathcal{K}_{-j})\big)\right),$$

*where $\mathcal{K} = \{\mathbf{K}_1, \ldots, \mathbf{K}_d\}$ with $\mathbf{K}_i$ being the kernel matrix for feature $i$ only, and $E_q(\mathcal{K}_{-j}) = \frac{1}{q}\sum_{r=1}^q(-1)^{r-1}E_{q-r}(\mathcal{K}_{-j})\bigodot P_r(\mathcal{K}_{-j})$, is the ESPs with $P_r(\mathcal{K}) = \sum_{\mathbf{K}_i \in \mathcal{K}}\mathbf{K}_i^r$ being the element-wise degree-$r$ power sum.*

The Shapley values $\phi_j^{\text{HSIC}}$ allow us to allocate the overall dependence between $X$ and $Y$ across individual features, identifying the most influential ones for prediction. Our results can be generalized to scenarios when both $X$ and $Y$ are multivariate; see Appendix G for more details. The attribution of overall dependence to the involved multivariate variables has other applications in statistical inference, e.g., kernel-based (conditional) independence testing (Gretton et al., 2007; Albert et al., 2022).

## 5 EXPERIMENTS

We evaluate the effectiveness of PKeX-Shapley for product-kernel methods through a series of experiments conducted on a 24-core machine with 16GB RAM. We further provide the experimental setups, training procedure, and extra experiments in Appendix H.

### 5.1 EFFECTIVENESS OF RECURSION AND VALUE FUNCTION IN LOCAL EXPLANATIONS

**Experiment 1: Recursion vs. Regression Formulation.** We empirically demonstrate the advantage of PKeX-Shapley, compared to sampling-based approximations such as Kernel SHAP. We generate four synthetic datasets, each with 1000 samples and $d \in \{10, 20, 30, 50\}$ features. Features are independently drawn from a standard normal distribution, and labels are generated using a linear model with additive Gaussian noise ($\sigma = 0.1$). For each dataset, we train a support vector regression model with an RBF kernel and use the trained model to compare explanation methods.

For a fair comparison between the recursive and regression-based methods, we adopt the same functional baseline value function as in equation 3. Specifically, we first compute the exact Shapley values $\phi_j^{\boldsymbol{x}}$ using PKeX-Shapley, and then estimate the Shapley values using a regression-based approach analogous to Kernel SHAP, but modified to employ our value function. The estimator uses

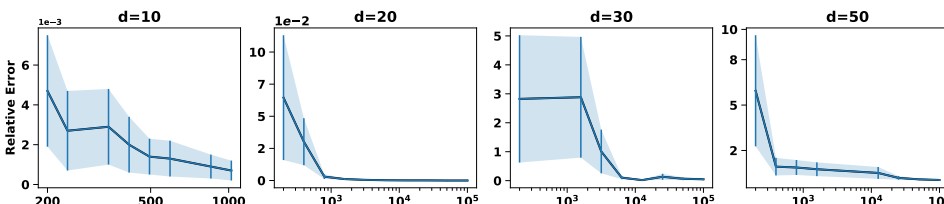

Figure 2: Relative estimation error of regression-based Shapley values versus the exact recursive values, shown across coalition sample sizes for feature dimensions $d = 10, 20, 30, 50$.

the paired coalition sampling scheme of Covert and Lee (2021), with sample sizes ranging from 200 to $10^5$, and computes $\hat{\phi}_j^{\boldsymbol{x}}$ by solving the corresponding weighted linear regression problem. To quantify the approximation error introduced by the regression formulation, we report the relative deviation error across 100 randomly selected instances, defined as $\sum_{j=1}^d |\phi_j^{\boldsymbol{x}} - \hat{\phi}_j^{\boldsymbol{x}}|/|\phi_j^{\boldsymbol{x}}|$.

Figure 2 plots the relative error as a function of sample size for each dataset over the 100 selected instances. When $d = 10$, the relative error is already around 0.005 with 200 coalition samples ($\approx$ 20% of all possible coalitions). However, when $d = 50$, the approximation error remains above 9.0 with 200 samples, stays above 1.0 with $10^4$, and only approaches 0.05 near $10^5$ coalition samples. These results highlight that as the number of features increases, a substantially larger number of samples is required to obtain reliable estimates—rendering sampling-based methods unreliable in high-dimensional settings. This, in turn, supports the importance of having a fast and exact computation algorithm.

**Experiment 2: Effectiveness of the Functional Baseline Value Function.** Recall that PKeX-Shapley employs a functional baseline value function, in contrast to standard expectation-based value functions. It is therefore natural to ask whether this formulation meaningfully captures the importance of feature subsets. To evaluate this, we assess the quality of the resulting explanations in recovering the most informative feature on synthetic datasets, and compare PKeX-Shapley against alternative explanation methods. We generate three regression tasks of $n = 1000$ samples in $\mathbb{R}^{50}$, where only the first one-third of features (denoted by $\mathcal{S}$, $|\mathcal{S}| = 17$) drive the target, and the remaining 33 features are redundant. The three target functions over $\mathcal{S}$ are: a degree-5 polynomial, a degree-10 polynomial, and a squared-exponential response $y = \exp(\sum_{i \in \mathcal{S}} x_i^2)$. We train a support vector regressor with an RBF kernel on each dataset, and produce explanations using our exact recursive method alongside three baselines: RKHS-SHAP (Chau et al., 2022), GEMFIX (Mohammadi et al., 2025a), BiSHAP (Masoomi et al., 2021), and Sampling SHAP (Štrumbelj and Kononenko, 2014), each configured with 500 and 1000 coalition samples. All methods employ a fixed background set of 100 points to estimate their value functions.

Attribution accuracy is computed over 100 independent test instances by selecting the top-17 features returned by each method and measuring the fraction of true active features recovered. Figure 3 shows the average accuracy rate for each method across the three tasks. PKeX-Shapley achieves a competitive or superior performance in all cases, whereas Kernel SHAP, GEMFIX, and Sampling SHAP suffer accuracy degradation as the target function's complexity increases (most notably for the degree-10 polynomial and the exponential model).

We also measure per-instance explanation runtime. Figure 4 presents error-bar plots (mean $\pm$ standard deviation) of execution times in seconds. With 500 coalition samples, all methods incur comparable execution times. When the sample size increases to 1000, PKeX-Shapley remains significantly faster than the baselines, despite using the same background-sample budget of 100 for other methods. This demonstrates that PKeX-Shapley not only provides exact attributions but also outperforms sampling-based estimators in computational efficiency as coalition sample counts grow.

## 5.2 EXPLAINING DISTRIBUTION DISCREPANCY USING MMD WITH PKEX-SHAPLEY

To illustrate how PKeX-Shapley can explain distributional discrepancies measured by MMD, we conduct two synthetic experiments following the standard two-sample testing setup (Schrab, 2025). Our goal is to attribute the observed MMD between two distributions to individual input variables. We present one of the synthetic experiments below, with the remaining experiments provided in Appendix H.3. In all MMD experiments, we use the RBF kernel with the bandwidth selected via the

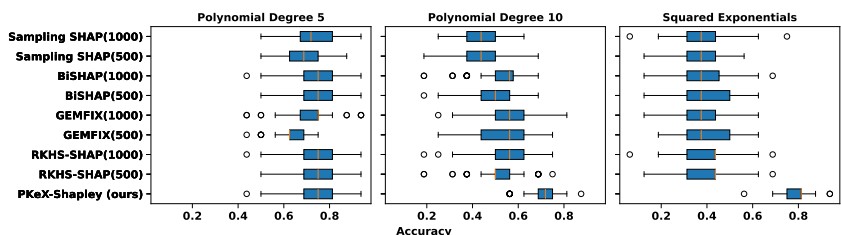

Figure 3: Recovery rate of true active features by each method on the three synthetic tasks.

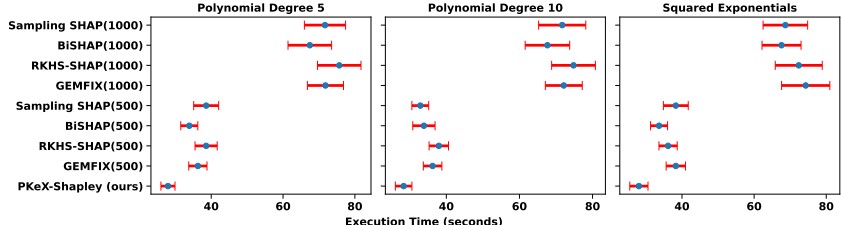

Figure 4: Per-instance explanation time (mean ± standard deviation) for each method with 500 and 1000 coalition samples.

median heuristic method. We generate datasets $X$ and $Z$, each comprising 20 variables. The first ten variables $X_1, \ldots, X_{10}$ and $Z_1, \ldots, Z_{10}$ are sampled from the same multivariate normal distribution, ensuring identical distributions. The remaining 10 variables $X_{11}, \ldots, X_{20}$ differ, with variables in $X$ sampled from a multivariate normal distribution and those in $Z$ from a Student's t-distribution of the same mean. This introduces differences in higher-order moments and covariance structure, resulting in a measurable discrepancy.

We generate 1,000 samples from each distribution and compute the MMD. Shapley values are computed to quantify the contribution of each variable to the overall MMD. To ensure robustness, the experiment was repeated 1,000 times, and kernel density estimates (KDE) of the Shapley values are presented in Figure 5.

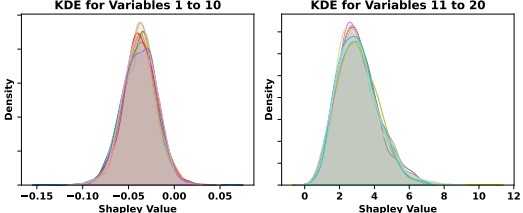

The left subplot in Figure 5 displays the KDE plots for variables 1–10, while the right subplot corresponds to variables 11–20. For the first 10 variables, the consistently negative Shapley values indicate a reduction in the overall MMD, ef-

Figure 5: KDE plots of Shapley values for the synthetic dataset. Variables 1–10 reduce MMD via negative contributions, while 11–20 increase it via positive contributions.

fectively "pulling" the discrepancy closer to zero. This aligns with our intuition: since these variables are identically distributed across both datasets, they should not contribute positively to the observed distributional difference. In contrast, variables 11–20 exhibit positive Shapley values, reflecting their contribution to the increase in MMD and, thus, their role in capturing the divergence between the distributions. Moreover, the KDE plots of the Shapley values within each group (variables 1–10 and 11–20, respectively) are nearly identical, which is consistent with the symmetric construction of the synthetic data. This experiment illustrates that Shapley values provide meaningful insights into the contribution of individual variables to distributional discrepancies measured by MMD.

### 5.3 EXPLAINING HSIC WITH PKEX-SHAPLEY: FEATURE SELECTION CASE STUDY

Lastly, we demonstrate how attributing the HSIC between input features and the target variable to individual features can support feature selection by quantifying their respective contributions to the overall statistical dependence. As a comparison, we also compare our approach with five feature importance methods: HSICLasso (Climente-González et al., 2019), Mutual Information (MI), Lasso, K–Best, and Tree Ensemble (with feature permutations). Our experiments were conducted on seven

Table 1: Performance (mean±standard deviation) when training on the top 20% of features. Datasets *breast cancer*, *skillcraft*, *sml*, and *parkinson* are regression (MAPE, lower is better); *sonar*, *Wisconsin*, *ionosphere* are classification (accuracy, higher is better).

| Method | sonar | Wisconsin | ionosphere | breast cancer | skillcraft | sml | parkinson |
|---|---|---|---|---|---|---|---|
| | accuracy (↑) | | | mean absolute percentage error (↓) | | | |
| PKeX-Shapley | 0.808±0.030 | **0.909±0.015** | 0.878±0.036 | **0.850±0.010** | **1.000±0.020** | 0.999±0.001 | **0.125±0.022** |
| HSICLasso | 0.808±0.044 | 0.884±0.010 | 0.912±0.035 | 1.000±0.080 | 2.175±0.429 | 1.354±0.726 | 1.170±0.269 |
| MI | **0.875±0.053** | 0.900±0.015 | **0.937±0.023** | 1.000±0.080 | 1.134±0.099 | **0.196±0.071** | 0.214±0.004 |
| Lasso | 0.842±0.038 | 0.900±0.024 | 0.932±0.021 | 1.000±0.080 | 1.821±0.680 | 1.000±0.000 | 1.000±0.000 |
| K–Best | 0.779±0.040 | **0.909±0.015** | 0.869±0.018 | 1.000±0.080 | 1.134±0.099 | 0.257±0.086 | 1.018±0.076 |
| Tree Ens. | 0.837±0.059 | 0.887±0.033 | 0.926±0.031 | 1.000±0.080 | 2.175±0.429 | 1.000±0.000 | 0.214±0.004 |

datasets, where we trained Gaussian process (GP) models with an RBF kernel, using only the top 20% of features ranked by each selection method.

Table 1 presents the results, reporting the five-fold cross-validated mean and standard deviation for the GP models trained on the selected features. We use mean absolute percentage error (MAPE) as the performance metric for regression tasks and accuracy for classification tasks. For kernel computation in HSIC, we use an RBF kernel for features and regression targets, and a categorical kernel for classification targets. The bandwidth for the RBF kernel is selected using the median heuristic. PKeX-Shapley consistently delivers strong results across all datasets. On regression problems, it yields better MAPE on datasets *breast cancer*, *skillcraft*, and *parkinson*, while other methods often incur higher error (e.g., HSICLasso on *skillcraft* and parkinson) or greater variance (e.g., HSICLasso on *sml*). For classification, PKeX-Shapley maintains accuracies above 80%, matching or exceeding the baselines. It achieves the best-performing results on the *Wisconsin* datasets, and remains competitive on the other two datasets. Interestingly, when compared to HSICLasso—a method specifically tailored for feature selection—PKeX-Shapley demonstrates superior performance across the majority of datasets. Notably, we achieve better results in 5 out of the 7 datasets, tie in one, and only fall short in the ionosphere dataset. This is particularly noteworthy, as PKeX-Shapley is not explicitly designed for feature selection, yet it consistently outperforms a specialized method like HSICLasso.

## 6 Conclusion, limitation, and discussion

This work introduces PKeX-Shapley, a polynomial-time algorithm for computing exact Shapley values for product-kernel methods. We introduce the functional baseline value function for product-kernel methods, and show that it naturally induces a functional decomposition, which we exploit to develop a recursive algorithm that bypass the exponential complexity of näive Shapley value computation. This approach enables exact, efficient feature attribution for both predictive models and kernel-based statistical discrepancies, including Maximum Mean Discrepancy (MMD) and Hilbert-Schmidt Independence Criterion (HSIC), providing interpretability of distributional differences and dependence structures. Our method achieves quadratic-time complexity and eliminates the approximation errors inherent in sampling-based estimators (and expectation-based value functions), as demonstrated through experiments on both synthetic and real-world datasets.

While our algorithm reduces computational complexity from exponential to quadratic time, its main limitation is that it applies only to product kernels. This trade-off is largely unavoidable: achieving tractable computation requires imposing structural constraints. As part of future work, we plan to explore whether further relaxation is possible. Another promising direction is to extend our approach to higher-order attribution methods, such as Shapley interaction indices (Sundararajan et al., 2020; Muschalik et al., 2024). It is also of interest to investigate how our computational techniques can speed up other explanation techniques, such as partial dependence plots and integrated gradients.

## Reproducibility Statement

We have made all code and implementation details publicly available (pke, 2025). The experimental setup, including dataset preprocessing, model architectures, training procedures, and hyperparameter settings, are fully described in either the main paper or the appendix. This ensures that all reported results can be independently verified.

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

APPENDIX

The appendix provides additional information and proofs related to the material presented in the main paper. It includes detailed explanations, proofs, algorithms, and experiments relevant to explaining product-kernel models. The structure of the appendix is as follows:

## A  PRODUCT KERNELS IN KERNEL METHODS

Product kernels provide a powerful mechanism for constructing high-dimensional similarity measures by combining kernels defined on individual dimensions or feature subsets. This section discusses key examples of product kernels.

**Radial Basis Function (RBF) Kernels as Product Kernels**  The RBF kernel is a canonical example of product kernels. The RBF kernel is defined based on a distance metric between two instances, with the two well-known metrics being Euclidean (L2 norm) and Manhattan distances (L1 norm). We refer to the former as RBF, and the latter as Laplacian RBF to distinguish these two kernel functions. In addition, when we have only one kernel bandwidth parameter $\sigma$, the RBF kernel is referred to as *isotropic*. The RBF kernel with both distance metrics inherently decomposes into products of univariate kernels across dimensions:

- **RBF Kernel**:

$$K_{\mathrm{RBF}}(\boldsymbol{x}, \boldsymbol{z}) = \exp\left(-\frac{\|\boldsymbol{x} - \boldsymbol{z}\|^2}{2\sigma^2}\right) = \prod_{i=1}^{d} \exp\left(-\frac{(x_i - z_i)^2}{2\sigma^2}\right).$$

- **Laplacian RBF Kernel**:

$$K_{\mathrm{Laplacian\,RBF}}(\boldsymbol{x}, \boldsymbol{z}) = \exp\left(-\frac{\|\boldsymbol{x} - \boldsymbol{z}\|_1}{\sigma}\right) = \prod_{i=1}^{d} \exp\left(-\frac{|x_i - z_i|}{\sigma}\right).$$

When alternative distance metrics are incorporated into the RBF kernel, such as the Mahalanobis distance, which involves a covariance matrix, the resulting kernel might lose its product decomposition.

**Automatic Relevance Determination (ARD) in Gaussian Processes**  ARD extends RBF kernels by assigning independent length-scale parameters $\sigma_i$ to each dimension:

$$K_{\mathrm{ARD}}(\boldsymbol{x}, \boldsymbol{z}) = \exp\left(-\sum_{i=1}^{d} \frac{(x_i - z_i)^2}{2\sigma_i^2}\right) = \prod_{i=1}^{d} \exp\left(-\frac{(x_i - z_i)^2}{2\sigma_i^2}\right).$$

The ARD is extensively used in Gaussian processes for feature selection via learned $\sigma_i$, and to enhance interpretability and adaptability. ARD is also referred to as *anisotropic*.

**Cauchy Kernel** The Cauchy kernel, inspired by the Cauchy distribution, is another example of a product kernel:

$$K_{\text{Cauchy}}(\boldsymbol{x}, \boldsymbol{z}) = \prod_{i=1}^{d} \frac{1}{1 + \frac{(x_i - z_i)^2}{\sigma^2}}.$$

**Product of Base Kernels** A popular way for constructing product kernels is by first defining a base kernel for each individual feature and then computing the overall kernel function by multiplying the base kernels over individual features. The product of PSD kernels remains PSD by the Schur product theorem:

$$K(\boldsymbol{x}, \boldsymbol{z}) = \prod_{i=1}^{d} K_i(x_i, z_i).$$

This type of kernel introduces flexibility in combining base kernels while maintaining validity as a (product) kernel function.

## B  VALUE FUNCTIONS FOR PRODUCT KERNELS: BASELINE AND INTERVENTIONAL

We now discuss the interventional and baseline value functions for product-kernel methods and their relevance and affinity with the proposed value function. For a product-kernel model, the decision function can be expressed as

$$f(\boldsymbol{x}) = \boldsymbol{\alpha}^\top \big( k_{\mathcal{S}}(\mathbf{X}_{\mathcal{S}}, \boldsymbol{x}_{\mathcal{S}}) \odot k_{\mathcal{S}^c}(\mathbf{X}_{\mathcal{S}^c}, \boldsymbol{x}_{\mathcal{S}^c}) \big),$$

where $\odot$ denotes the Hadamard product, $k_{\mathcal{S}}$ is the product kernel restricted to subset $\mathcal{S}$, and $\mathcal{S}^c$ is its complement.

### B.1  BASELINE VALUE FUNCTION

The baseline value function removes the effect of features outside the coalition $\mathcal{S}$ by replacing them with fixed baseline values. Formally, let $\boldsymbol{x}_{S^c}^{b}$ denote the baseline values for the excluded features. Then the baseline value function is defined as

$$\begin{aligned}
v_{\boldsymbol{x}}^{\text{base}}(\mathcal{S}) &= f([\boldsymbol{x}_{\mathcal{S}}, \boldsymbol{x}_{S^c}^{b}]) \\
&= \boldsymbol{\alpha}^\top \mathbf{K}([\boldsymbol{x}_{\mathcal{S}}, \boldsymbol{x}_{S^c}^{b}], \mathbf{X}) = \boldsymbol{\alpha}^\top \big( k_{\mathcal{S}}(\mathbf{X}_{\mathcal{S}}, \boldsymbol{x}_{\mathcal{S}}) \odot k_{\mathcal{S}^c}(\mathbf{X}_{\mathcal{S}^c}, \boldsymbol{x}_{\mathcal{S}^c}^{b}) \big).
\end{aligned}$$

In model-agnostic settings, the choice of baseline values is often application-dependent (e.g., mean input, zero vector, or reference sample). For product-kernel methods, however, we have an explicit multiplicative structure. Instead of fixing raw feature values, we can directly mask the kernel factors corresponding to excluded features. Specifically, setting a kernel factor to one corresponds to the least informative contribution, since it makes the kernel computation independent of that feature. Thus, the baseline operation in product kernels amounts to replacing $k_j(x_j, x_j^{(i)})$ with 1 for all $j \in S^c$. This yields

$$v_{\boldsymbol{x}}(\mathcal{S}) = \boldsymbol{\alpha}^\top k_{\mathcal{S}}(\mathbf{X}_{\mathcal{S}}, \boldsymbol{x}_{\mathcal{S}}),$$

which is exactly the *functional baseline value function*.

In words, *rather than choosing baseline feature values in input space, product-kernel methods define the baseline at the level of the kernel: excluded features are masked by setting their kernel contributions to* 1. This ensures that only features in $\mathcal{S}$ influence the similarity measure.

### B.2  INTERVENTIONAL VALUE FUNCTION

An alternative approach is the *interventional value function*, which removes the effect of features outside $\mathcal{S}$ by replacing them with their *average contribution* under the data distribution. Formally, it

is defined as

$$v_{\boldsymbol{x}}^{\text{int}}(\mathcal{S}) = \mathbb{E}_{\boldsymbol{x}_{\mathcal{S}^c}}\left[f(\boldsymbol{x}) \mid \boldsymbol{x}_{\mathcal{S}}\right]$$
$$= \mathbb{E}_{\boldsymbol{x}_{\mathcal{S}^c}}\left[\boldsymbol{\alpha}^\top k([\boldsymbol{x}_{\mathcal{S}}, \boldsymbol{x}_{\mathcal{S}^c}], \mathbf{X})\right]$$
$$= \boldsymbol{\alpha}^\top \mathbb{E}_{\boldsymbol{x}_{\mathcal{S}^c}}\left[k([\boldsymbol{x}_{\mathcal{S}}, \boldsymbol{x}_{\mathcal{S}^c}], \mathbf{X})\right].$$

Exploiting the product structure of the kernel, this simplifies to

$$v_{\boldsymbol{x}}^{\text{int}}(\mathcal{S}) = \boldsymbol{\alpha}^\top \left(k_{\mathcal{S}}(\boldsymbol{x}_{\mathcal{S}}, \mathbf{X}_{\mathcal{S}}) \odot \mathbb{E}_{\boldsymbol{x}_{\mathcal{S}^c}}\left[k_{\mathcal{S}^c}(\boldsymbol{x}_{\mathcal{S}^c}, \mathbf{X}_{\mathcal{S}^c})\right]\right). \tag{5}$$

To summarize, *the interventional value function removes a feature's contribution by averaging it out, while our baseline-inspired value function directly eliminates the feature's influence by setting its kernel component to one.*

This highlights the conceptual and practical difference: the interventional function is defined through expectations with respect to the data distribution, whereas our proposed value function exploits the explicit multiplicative decomposition of product kernels and provides an efficient, distribution-free alternative.

## C   NEWTON'S IDENTITIES

To explain Newton's identities, we begin with a specific set of variables $\mathcal{Z}_4 = \{z_1, z_2, z_3, z_4\}$ before generalizing it to sets of arbitrary size $\mathcal{Z}_d = \{z_1, z_2, \ldots, z_d\}$. The *elementary symmetric polynomials* (ESPs) of degree $q$ is defined as

$$e_q(\mathcal{Z}_4) = \sum_{1 \leq i_1 < i_2 < \cdots < i_q \leq 4} z_{i_1} z_{i_2} \cdots z_{i_q},$$

with the conventions $e_0(\mathcal{Z}_4) = 1$ and $e_q(\mathcal{Z}_4) = 0$ for $q > 4$. For example:

$$e_1(\mathcal{Z}_4) = z_1 + z_2 + z_3 + z_4,$$
$$e_2(\mathcal{Z}_4) = z_1 z_2 + z_1 z_3 + z_1 z_4 + z_2 z_3 + z_2 z_4 + z_3 z_4,$$
$$e_4(\mathcal{Z}_4) = z_1 z_2 z_3 z_4.$$

The *power sum* of degree $r$ is given by

$$p_r(\mathcal{Z}_4) = z_1^r + z_2^r + z_3^r + z_4^r.$$

In particular, $p_1(\mathcal{Z}_4) = e_1(\mathcal{Z}_4)$ and, for example, $p_2(\mathcal{Z}_4) = z_1^2 + z_2^2 + z_3^2 + z_4^2$. Then, Newton's identities relate the ESPs to the power sum recursively. For $q \geq 1$,

$$e_q(\mathcal{Z}_4) = \frac{1}{q} \sum_{r=1}^{q} (-1)^{r-1} e_{q-r}(\mathcal{Z}_4)\, p_r(\mathcal{Z}_4).$$

For the set $\mathcal{Z}_4 = \{z_1, z_2, z_3, z_4\}$, the identities yield:

$$e_1(\mathcal{Z}_4) = \frac{1}{1}[e_0(\mathcal{Z}_4)\, p_1(\mathcal{Z}_4)] = p_1(\mathcal{Z}_4) = z_1 + z_2 + z_3 + z_4,$$

$$e_2(\mathcal{Z}_4) = \frac{1}{2}\Big[e_1(\mathcal{Z}_4)\, p_1(\mathcal{Z}_4) - e_0(\mathcal{Z}_4)\, p_2(\mathcal{Z}_4)\Big] = \frac{(z_1 + z_2 + z_3 + z_4)^2 - (z_1^2 + z_2^2 + z_3^2 + z_4^2)}{2},$$

$$e_3(\mathcal{Z}_4) = \frac{1}{3}\Big[e_2(\mathcal{Z}_4)\, p_1(\mathcal{Z}_4) - e_1(\mathcal{Z}_4)\, p_2(\mathcal{Z}_4) + e_0(\mathcal{Z}_4)\, p_3(\mathcal{Z}_4)\Big],$$

$$e_4(\mathcal{Z}_4) = \frac{1}{4}\Big[e_3(\mathcal{Z}_4)\, p_1(\mathcal{Z}_4) - e_2(\mathcal{Z}_4)\, p_2(\mathcal{Z}_4) + e_1(\mathcal{Z}_4)\, p_3(\mathcal{Z}_4) - e_0(\mathcal{Z}_4)\, p_4(\mathcal{Z}_4)\Big].$$

The identities presented above can be extended to sets of arbitrary size $\mathcal{Z}_d = \{z_1, z_2, \ldots, z_d\}$ as

$$e_q(\mathcal{Z}_d) = \frac{1}{q} \sum_{r=1}^{q} (-1)^{r-1} e_{q-r}(\mathcal{Z}_d)\, p_r(\mathcal{Z}_d), \quad \text{for } q \geq 1,$$

with $e_0(\mathcal{Z}_d) = 1$ and $e_q(\mathcal{Z}_d) = 0$ if $q > d$ or $q < 0$.

# D PROOFS

This section provides the detailed proofs of the theoretical results presented in the main paper. First of all, we present a theorem that plays a key role in the other proofs.

## D.1 PROOF OF THEOREM 2

We can write the Shapley value as follows:

$$\phi_j^{\boldsymbol{x}} := \sum_{\mathcal{S} \subseteq \mathcal{D} \setminus \{j\}} \mu(|\mathcal{S}|)\big(v_{\boldsymbol{x}}(\mathcal{S} \cup \{j\}) - v_{\boldsymbol{x}}(\mathcal{S})\big) = \sum_{q=0}^{d-1} \mu(q) \sum_{\substack{\mathcal{S} \subseteq \mathcal{D} \setminus \{j\} \\ |\mathcal{S}| = q}} \bigg( v_{\boldsymbol{x}}(\mathcal{S} \cup \{j\}) - v_{\boldsymbol{x}}(\mathcal{S}) \bigg).$$

We now substitute $v_{\boldsymbol{x}}(\mathcal{S})$ for the product kernel into the Shapley value formula, one gets:

$$\phi_j^{\boldsymbol{x}} = \boldsymbol{\alpha}^\top \left( \sum_{q=0}^{d-1} \mu(q) \sum_{\substack{\mathcal{S} \subseteq \mathcal{D} \setminus \{j\} \\ |\mathcal{S}| = q}} k_{\mathcal{S} \cup \{j\}}(\mathbf{X}_{\mathcal{S} \cup \{j\}}, \boldsymbol{x}_{\mathcal{S} \cup \{j\}}) - k_{\mathcal{S}}(\mathbf{X}_{\mathcal{S}}, \boldsymbol{x}_{\mathcal{S}}) \right)$$

$$= \boldsymbol{\alpha}^\top \left( \sum_{q=0}^{d-1} \mu(q) \sum_{\substack{\mathcal{S} \subseteq \mathcal{D} \setminus \{j\} \\ |\mathcal{S}| = q}} k_j(\mathbf{X}_j, x_j) \odot k_{\mathcal{S}}(\mathbf{X}_{\mathcal{S}}, \boldsymbol{x}_{\mathcal{S}}) - k_{\mathcal{S}}(\mathbf{X}_{\mathcal{S}}, \boldsymbol{x}_{\mathcal{S}}) \right)$$

$$= \boldsymbol{\alpha}^\top \left( \Big( k_j(\mathbf{X}_j, x_j) - \mathbf{1} \Big) \odot \left( \sum_{q=0}^{d-1} \mu(q) \sum_{\substack{\mathcal{S} \subseteq \mathcal{D} \setminus \{j\} \\ |\mathcal{S}| = q}} k_{\mathcal{S}}(\mathbf{X}_{\mathcal{S}}, \boldsymbol{x}_{\mathcal{S}}) \right) \right), \tag{6}$$

where $\mathbf{1}$ is a vector of one. Let $\boldsymbol{z}_i = k_i(\mathbf{X}_i, x_i)$ and $\mathcal{Z} = \{\boldsymbol{z}_1, ..., \boldsymbol{z}_d\}$, one can write:

$$\sum_{\substack{\mathcal{S} \subseteq \mathcal{D} \setminus \{j\} \\ |\mathcal{S}| = q}} k_{\mathcal{S}}(\mathbf{X}, \boldsymbol{x}) = \sum_{\substack{\mathcal{S} \subseteq \mathcal{D} \setminus \{j\} \\ |\mathcal{S}| = q}} \bigodot_{i \in \mathcal{S}} k_i(\mathbf{X}, \boldsymbol{x}) = \sum_{\substack{1 \leq i_1 < i_2 < \cdots < i_q \leq d-1 \\ j \notin \{i_1, ..., i_q\}}} \boldsymbol{z}_{i_1} \odot \boldsymbol{z}_{i_2} \cdots \odot \boldsymbol{z}_{i_q} = e_q(\mathcal{Z}_{-j}),$$

where $e_q(\mathcal{Z}_{-j})$ is the elementary symmetric polynomials. This equation means that the inner sum can be recursively computed by Newton's identities formulation. It then follows:

$$\phi_j^{\boldsymbol{x}} = \boldsymbol{\alpha}^\top \left( \Big( k_j(\mathbf{X}_j, x_j) - \mathbf{1} \Big) \odot \sum_{q=0}^{d-1} \mu(q) e_q(\mathcal{Z}_{-j}) \right)$$

where

$$e_q(\mathcal{Z}_{-j}) = \frac{1}{q} \sum_{r=1}^{q} (-1)^{r-1} e_{q-r}(\mathcal{Z}_{-j}) \odot p_r(\mathcal{Z}_{-j}),$$

and $p_r(\mathcal{Z}) = \sum_{\boldsymbol{z}_i \in \mathcal{Z}} \boldsymbol{z}_i^r$ is the power sum, with the power working element-wise. This completes the proof. $\qquad\square$

## D.2 PROOF OF LEMMA 3

By the efficiency property of Shapley values (Shapley, 1953), the sum of Shapley values equals the difference between the value of the grand coalition and the empty coalition:

$$\sum_{j=1}^{d} \phi_j^{\boldsymbol{x}} = v_{\boldsymbol{x}}(\mathcal{D}) - v_{\boldsymbol{x}}(\emptyset).$$

For the functional baseline value function, we have:

$$v_{\boldsymbol{x}}(\mathcal{D}) = \boldsymbol{\alpha}^\top k_{\mathcal{D}}(\mathbf{X}_{\mathcal{D}}, \boldsymbol{x}_{\mathcal{D}}) = f(\boldsymbol{x}),$$

$$v_{\boldsymbol{x}}(\emptyset) = \boldsymbol{\alpha}^\top k_{\emptyset}(\mathbf{X}_{\emptyset}, \boldsymbol{x}_{\emptyset}) = \boldsymbol{\alpha}^\top \mathbf{1} = \sum_{i=1}^{n} \alpha_i = f_{\emptyset}(\boldsymbol{x}).$$

The result follows immediately by substitution.

### D.3 PROOF OF PROPOSITION 5

By substituting $v_{\text{MMD}}(\mathcal{S})$ into the Shapley value formula, we obtain:

$$\phi_q^{\text{MMD}} = \sum_{r=0}^{d-1} \mu(r) \sum_{\substack{\mathcal{S} \subseteq \mathcal{D}\setminus\{q\} \\ |\mathcal{S}|=r}} \left( v_{\text{MMD}}(\mathcal{S} \cup \{q\}) - v_{\text{MMD}}(\mathcal{S}) \right)$$

$$= \sum_{r=0}^{d-1} \mu(r) \sum_{\substack{\mathcal{S} \subseteq \mathcal{D}\setminus\{q\} \\ |\mathcal{S}|=r}} \left( \frac{1}{n(n-1)} \sum_{i \neq j} \left( k_q(x_q^{(i)}, x_q^{(j)}) - 1 \right) k_{\mathcal{S}}(\boldsymbol{x}_{\mathcal{S}}^{(i)}, \boldsymbol{x}_{\mathcal{S}}^{(j)}) \right.$$

$$+ \frac{1}{m(m-1)} \sum_{i \neq j} \left( k_q(z_q^{(i)}, z_q^{(j)}) - 1 \right) k_{\mathcal{S}}(\boldsymbol{z}_{\mathcal{S}}^{(i)}, \boldsymbol{z}_{\mathcal{S}}^{(j)})$$

$$\left. - \frac{2}{nm} \sum_{i,j} \left( k_q(x_q^{(i)}, z_q^{(j)}) - 1 \right) k_{\mathcal{S}}(\boldsymbol{x}_{\mathcal{S}}^{(i)}, \boldsymbol{z}_{\mathcal{S}}^{(j)}) \right).$$

Let $\mathcal{Z}^{(\boldsymbol{x}^{(i)}, \boldsymbol{x}^{(j)})} = \{k_1(x_1^{(i)}, x_1^{(j)})), \ldots, k_d(x_d^{(i)}, x_d^{(j)}))\}$ and define the elementary symmetric polynomial as

$$e_r(\mathcal{Z}_{-q}^{(\boldsymbol{x}^{(i)}, \boldsymbol{x}^{(j)})}) = \sum_{\substack{\mathcal{S} \subseteq \mathcal{D}\setminus\{q\} \\ |\mathcal{S}|=r}} k_{\mathcal{S}}(\boldsymbol{x}_{\mathcal{S}}^{(i)}, \boldsymbol{x}_{\mathcal{S}}^{(j)}).$$

Then, it follows from Newton's identities recurrence:

$$e_r(\mathcal{Z}_{-q}^{(\boldsymbol{x}^{(i)}, \boldsymbol{x}^{(j)})}) = \frac{1}{r} \sum_{s=1}^{r} (-1)^{s-1} e_{r-s}(\mathcal{Z}_{-q}^{(\boldsymbol{x}^{(i)}, \boldsymbol{x}^{(j)})}) p_s(\mathcal{Z}_{-q}^{(\boldsymbol{x}^{(i)}, \boldsymbol{x}^{(j)})}),$$

where $p_s(\mathcal{Z}) = \sum_{z \in \mathcal{Z}} z^s$. Finally, we obtain

$$\phi_q^{\text{MMD}} = \frac{1}{n(n-1)} \sum_{i \neq j} \left( \left( k_q(x_q^{(i)}, x_q^{(j)}) - 1 \right) \sum_{r=0}^{d-1} \mu(r) e_r(\mathcal{Z}_{-q}^{(x_q^{(i)}, x_q^{(j)})}) \right)$$

$$+ \frac{1}{m(m-1)} \sum_{i \neq j} \left( \left( k_q(z_q^{(i)}, z_q^{(j)}) - 1 \right) \sum_{r=0}^{d-1} \mu(r) e_r(\mathcal{Z}_{-q}^{(\boldsymbol{z}^{(i)}, \boldsymbol{z}^{(j)})}) \right)$$

$$- \frac{2}{nm} \sum_{i,j} \left( \left( k_q(x_q^{(i)}, z_q^{(j)}) - 1 \right) \sum_{r=0}^{d-1} \mu(r) e_r(\mathcal{Z}_{-q}^{(\boldsymbol{x}^{(i)}, \boldsymbol{z}^{(j)})}) \right),$$

and that completes the proof. $\qquad\square$

### D.4 PROOF OF PROPOSITION 7

By substituting $v_{\text{HSIC}}(\mathcal{S}) = \frac{1}{(n-1)^2} \text{tr}(\mathbf{K}_{\mathcal{S}} \mathbf{H} \mathbf{L} \mathbf{H})$ into the Shapley value formula, we obtain:

$$\phi_j^{\text{HSIC}} = \frac{1}{(n-1)^2} \text{tr} \left( \mathbf{H} \mathbf{L} \mathbf{H} \sum_{q=0}^{d-1} \mu(q) \sum_{\substack{\mathcal{S} \subseteq \mathcal{D}\setminus\{j\} \\ |\mathcal{S}|=q}} \left( \mathbf{K}_{\mathcal{S} \cup \{j\}} - \mathbf{K}_{\mathcal{S}} \right) \right)$$

$$= \frac{1}{(n-1)^2} \text{tr} \left( \mathbf{H} \mathbf{L} \mathbf{H} \sum_{q=0}^{d-1} \mu(q) \sum_{\substack{\mathcal{S} \subseteq \mathcal{D}\setminus\{j\} \\ |\mathcal{S}|=q}} \left( \mathbf{K}_j \odot \mathbf{K}_{\mathcal{S}} - \mathbf{K}_{\mathcal{S}} \right) \right)$$

$$= \frac{1}{(n-1)^2} \text{tr} \left( \mathbf{H} \mathbf{L} \mathbf{H} (\mathbf{K}_j - \mathbf{1} \mathbf{1}^{\top}) \odot \sum_{q=0}^{d-1} \mu(q) \sum_{\substack{\mathcal{S} \subseteq \mathcal{D}\setminus\{j\} \\ |\mathcal{S}|=q}} \mathbf{K}_{\mathcal{S}} \right).$$

Letting $\mathcal{K} = \{\mathbf{K}_1, ..., \mathbf{K}_d\}$, we express:

$$\sum_{\substack{\mathcal{S} \subseteq \mathcal{D} \setminus \{j\} \\ |\mathcal{S}| = q}} \mathbf{K}_\mathcal{S} = E_q(\mathcal{K}_{-j}),$$

which is computed recursively via Newton's identities formulation:

$$E_q(\mathcal{K}_{-j}) = \frac{1}{d-1} \sum_{r=1}^{d-1} (-1)^{r-1} E_{q-r}(\mathcal{K}_{-j}) \odot P_r(\mathcal{K}_{-j}),$$

where $P_r(\mathcal{K}) = \sum_{\mathbf{K}_i \in \mathcal{K}} \mathbf{K}_i^r$ is the element-wise power sum polynomial. Substituting this back, we obtain:

$$\phi_j^{\text{HSIC}} = \frac{1}{(n-1)^2} \text{tr} \left( \mathbf{HLH} \Big( (\mathbf{K}_j - \mathbf{1}\mathbf{1}^\top) \bigodot \sum_{q=0}^{d-1} \mu(q) E_q(\mathcal{K}_{-j}) \Big) \right),$$

which completes the proof. $\qquad\square$

# E   RECURSIVE AND NUMERICALLY STABLE ALGORITHMS FOR COMPUTING SHAPLEY VALUES FOR PRODUCT-KERNEL LEARNING MODELS

---

**Algorithm 1** Recursive Computaiton of Shapley Values for Product-Kernel Learning Models

---

**Require:** Trained model with product kernels (SVM/SVR/GP), instance $\boldsymbol{x} \in \mathbb{R}^d$
**Ensure:** Shapley values $\phi_1^{\boldsymbol{x}}, \ldots, \phi_d^{\boldsymbol{x}}$ for each feature
 1: Retrieve training data $X$, coefficients $\alpha$, and kernel function $k$
 2: Compute feature-wise kernel vectors $\boldsymbol{z}_j = k(\boldsymbol{x}, X), \forall j \in \{1, \ldots, d\}$
 3: Precompute coefficients $\mu(q) = \frac{q!(d-q-1)!}{d!}, q = 0, \ldots, d-1$
 4: **for** $j \in \{1, \ldots, d\}$ **do**
 5:     Let $\mathcal{Z}_{-j} \leftarrow \{\boldsymbol{z}_1, \ldots, \boldsymbol{z}_d\} \setminus \{\boldsymbol{z}_j\}$
 6:     Compute power sums $p_r(\mathcal{Z}_{-j}) = \sum_{\boldsymbol{z} \in \mathcal{Z}_{-j}} \boldsymbol{z}^r$ for $r = 1, \ldots, d-1$
 7:     Initialize $e_0 \leftarrow \mathbf{1}$
 8:     **for** $q = 1$ **to** $d-1$ **do**
 9:         $e_q(\mathcal{Z}_{-j}) \leftarrow \frac{1}{q} \sum_{r=1}^{q} (-1)^{k-1} e_{q-r}(\mathcal{Z}_{-j}) \odot p_r(\mathcal{Z}_{-j})$
10:     **end for**
11:     $\psi_j \leftarrow \sum_{q=0}^{d-1} \mu(q) \cdot e_q(\mathcal{Z}_{-j})$
12:     $\phi_j^{\boldsymbol{x}} \leftarrow \boldsymbol{\alpha}^\top ((\boldsymbol{z}_j - \mathbf{1}) \odot \psi_j)$
13: **end for**
14:
15: **return** $(\phi_1^{\boldsymbol{x}}, \ldots, \phi_d^{\boldsymbol{x}})$

---

We present the algorithm for computing the Shapley values for product-kernel learning models. Let $\mathcal{Z} = \{\boldsymbol{z}_1, \ldots, \boldsymbol{z}_d\}$ be a collection of kernel vectors. The elementary symmetric polynomial (ESP) of degree $q$ is defined as:

$$e_q(\mathcal{Z}) = \sum_{1 \leq i_1 < \cdots < i_q \leq d} \boldsymbol{z}_{i_1} \odot \cdots \odot \boldsymbol{z}_{i_q}.$$

Traditional computation uses Newton's identities:

$$e_q(\mathcal{Z}) = \frac{1}{q} \sum_{k=1}^{q} (-1)^{k-1} e_{q-k}(\mathcal{Z}) p_k(\mathcal{Z}), \quad p_k(\mathcal{Z}) = \sum_{\boldsymbol{z}_i \in \mathcal{Z}} \boldsymbol{z}_i^k$$

We used this recursion to compute Shapley values in product-kernel learning models. Algorithm 1 summarizes the overall algorithm based on this recursive formulation.

---

**Algorithm 2** Numerically Stable Shapley Values Computation

---

**Require:** Trained model with product kernels (SVM/SVR/GP), instance $\boldsymbol{x} \in \mathbb{R}^d$
**Ensure:** Shapley values $\phi_1^{\boldsymbol{x}}, \ldots, \phi_d^{\boldsymbol{x}}$ for each feature
1: Retrieve training data $X$, coefficients $\boldsymbol{\alpha}$, and kernel function $k$
2: Compute feature-wise kernel vectors $\boldsymbol{z}_j = k(\boldsymbol{x}, X), \forall j \in \{1, \ldots, d\}$
3: Precompute coefficients $\mu(q) = \frac{q!(d-q-1)!}{d!}, q = 0, \ldots, d-1$
4: **for** $j = 1$ **to** $d$ **do**
5:     $\mathcal{Z}_{-j} \leftarrow \{\boldsymbol{z}_1, \ldots, \boldsymbol{z}_d\} \setminus \{\boldsymbol{z}_j\}$
6:     Scale $\mathcal{Z}_{-j}$: $s \leftarrow \max_{ij} \boldsymbol{z}_{ij} \forall \boldsymbol{z}_i \in \mathcal{Z}_{-j}$ (or 1 if all 0)
7:     $\tilde{\mathcal{Z}}_{-j} \leftarrow \{\boldsymbol{z}_i/s \mid \boldsymbol{z}_i \in \mathcal{Z}_{-j}\}$
8:     Coeff $\leftarrow [1] \{P_0(\boldsymbol{x}) = 1\}$
9:     **for** $\tilde{\boldsymbol{z}}_i \in \tilde{\mathcal{Z}}_{-j}$ **do**
10:       new_coeff $\leftarrow [\,]$
11:       **Append** $-\tilde{\boldsymbol{z}}_i \cdot$ Coeff[0] to new_coeff {Term for $\boldsymbol{x}^0$}
12:       **for** $k = 1$ **to** $\text{len}(\text{Coeff}) - 1$ **do**
13:         term $\leftarrow$ Coeff[$k-1$] $- (\tilde{\boldsymbol{z}}_i \odot \text{Coeff}[k])$
14:         **Append** term to new_coeff
15:       **end for**
16:       **Append** Coeff[$-1$] to new_coeff
17:       Coeff $\leftarrow$ new_coeff
18:     **end for**
19:     Extract $e_q(\mathcal{Z}_{-j})$: $e_q(\mathcal{Z}_{-j}) \leftarrow (-1)^q \cdot \text{Coeff}[d-q-1] \cdot s^q$ for $q = 0, \ldots, d-1$
20:     $\psi_j \leftarrow \sum_{q=0}^{d-1} \mu(q) \odot e_q(\mathcal{Z}_{-j})$
21:     $\phi_j^{\boldsymbol{x}} \leftarrow \boldsymbol{\alpha}^\top ((\boldsymbol{z}_j - \mathbf{1}) \odot \psi_j)$
22: **end for**
23: **return** $(\phi_1^{\boldsymbol{x}}, \ldots, \phi_d^{\boldsymbol{x}})$

---

Though being efficient, the recursive formulation suffers from numerical instability due to alternating sign cancellation, power operations, and error amplification through division. To develop a stable approach, we use the fact that symmetric elementary polynomials emerge naturally as coefficients of the characteristic polynomial (Egge, 2019):

$$P(\boldsymbol{x}) = \bigodot_{i=1}^{d}(\boldsymbol{x} - \boldsymbol{z}_i) = \sum_{q=0}^{d}(-1)^{d-q}e_{d-q}(\mathcal{Z})\boldsymbol{x}^q.$$

Initialize $P_0(\boldsymbol{x}) = \mathbf{1}$. For each $\boldsymbol{z}_i$, the update rule is

$$P_i(\boldsymbol{x}) = P_{i-1}(\boldsymbol{x}) \odot (\boldsymbol{x} - \boldsymbol{z}_i),$$

and the coefficients evolve as

$$\text{Coeff}_m^{(i)} = \text{Coeff}_{m-1}^{(i-1)} - \left(\boldsymbol{z}_i \bigodot \text{Coeff}_m^{(i-1)}\right),$$

where $\text{Coeff}_m^{(i)}$ denotes the coefficient of $\boldsymbol{x}^m$ after processing $i$ elements. The elementary symmetric polynomials are then:

$$e_q(\mathcal{Z}) = (-1)^{d-q} \cdot \text{Coeff}_q^{(d)}.$$

Using this relationship, we can avoid the power and division operations in the standard recursive formulation and develop a more stable algorithm to compute ESPs. Aside from this, we scale kernel vectors $\boldsymbol{z}_i \leftarrow \boldsymbol{z}_i/s$ where $s = \max_{ij} \boldsymbol{z}_{ij}$ to prevent overflow, with final correction $e_q \leftarrow e_q \cdot s^q$. Algorithm 2 shows the numerically stable algorithm for computing Shapley values for product-kernel learning models. This algorithm has the same complexity as Algorithm 1, but it avoids the power and division, which makes it more numerically stable (especially for high values of $d$).

Computing Shapley values for product-kernel models requires evaluating the marginal contribution of each feature across all possible coalitions, which naïvely involves computing $d$ different sets of ESPs for the leave-one-out collections $\mathcal{Z}_{-j}$. It makes the overall time-complexity $O(d^3)$ for all $d$ features. To overcome these limitations, we exploit a key structural insight: the ESPs of any subset $\mathcal{Z}_{-j}$ appear as coefficients of the characteristic polynomial obtained by removing a single linear factor. Thus, instead of recomputing ESPs independently, we construct the global polynomial $P(t) = \prod_{i=1}^{d}(t - \tilde{\boldsymbol{z}}_i)$ once, and then obtain each leave-one-out polynomial $Q^{(j)}(t) = P(t)/(t - \tilde{\boldsymbol{z}}_j)$ using synthetic division. This procedure yields all ESPs $e_q(\mathcal{Z}_{-j})$ for every feature $j$ at the cost of a single backward pass over the polynomial coefficients. The resulting algorithm is both numerically stable—avoiding powers, divisions, and alternating-sign recurrences—and achieves a quadratic $O(d^2)$ runtime, matching the efficiency of TreeSHAP while being fully compatible with product-kernel methods. In summary, the synthetic-division algorithm provides an exact, scalable, and stable method for Shapley value computation in multiplicative kernel models, enabling efficient instance-wise explainability without approximations.

The algorithm first constructs the global characteristic polynomial $P(t) = \prod_{j=1}^{d}(t - \tilde{\boldsymbol{z}}_j)$, whose coefficient array is obtained by successively multiplying by linear factors via synthetic division. At step $j$, the current polynomial has degree $j - 1$, so the update touches $O(j)$ coefficient vectors. Summing over $j = 1, \ldots, d$ yields $O(d^2)$ operations in the feature dimension. Next, for each feature $j$, we perform a single synthetic division of $P(t)$ by $(t - \tilde{\boldsymbol{z}}_j)$, requiring $O(d)$ operations and producing the leave-one-out polynomial $Q^{(j)}(t)$. Repeating this for all $d$ features again costs $O(d^2)$. Finally, extracting ESPs, forming the terms $\psi_j$, and computing each Shapley value $\phi_j^{\boldsymbol{x}}$ requires $O(d)$ per feature, again totalling $O(d^2)$. Overall, the full procedure computes all Shapley values in $O(d^2)$ time, while maintaining numerical stability and avoiding the cubic $O(d^3)$ cost of recomputing ESPs independently for each feature. Algorithm 3 summarizes the algorithm

# F  ADDITIVITY OF EXPLANATIONS FOR LEARNING MODELS, MMD AND HSIC

## F.1  EXPLANATION ADDITIVITY FOR LEARNING MODELS

In Lemma 3, we established the additivity property of the explanation in product-kernel learning models. In particular, we demonstrated that

$$f(\boldsymbol{x}) - \boldsymbol{\alpha}^\top \mathbf{1} = \sum_j \phi_j^{\boldsymbol{x}},$$

where $\boldsymbol{\alpha}^\top \mathbf{1}$ represents the value of the null game according to the value function in equation 3. This result is useful since $\boldsymbol{\alpha}^\top \mathbf{1} = 0$ for several kernel methods such as support vector machines (we have the constraint $\sum \hat{\alpha}_j y_j = 0$ in the dual problem and $f(\boldsymbol{x}) = \sum_j \hat{\alpha}_j y_j k(\boldsymbol{x}, \boldsymbol{x}_j)$ with $\hat{\boldsymbol{\alpha}}$ be the solution to the dual problem) and support vector regression. However, in general, it distributes the value of $f(\boldsymbol{x})$ only after subtracting the null game's value rather than allocating the full output $f(\boldsymbol{x})$ directly among the features. This is because $k_\emptyset = 1$ by definition, and this will lead to $v(\emptyset) = \boldsymbol{\alpha}^\top \mathbf{1}$. We now show that by redefining the value function in equation 3 so that $v(\emptyset) = 0$, the corresponding Shapley values will sum to $f(\boldsymbol{x})$, with each Shapley value augmented by $\frac{\boldsymbol{\alpha}^\top \mathbf{1}}{n}$.

**Proposition 8.** *Define a normalized value function by setting the kernel component for the empty set to zero, i.e., $k_\emptyset = 0$. Let $\hat{v}_{\boldsymbol{x}}$ be the corresponding value function and $\hat{\phi}_j$ the resulting Shapley values, which satisfy*

$$f(\boldsymbol{x}) = \sum_{j=1}^{n} \hat{\phi}_j.$$

*Then, assuming an equal allocation of the baseline value, the following relationship holds for every feature $j$:*

$$\hat{\phi}_j = \phi_j + \frac{\boldsymbol{\alpha}^\top \mathbf{1}}{n}.$$

---

**Algorithm 3** Shapley Values via Synthetic Division for Product-Kernel Methods

---

**Require:** Trained product-kernel model (SVM/SVR/GP), instance $\boldsymbol{x} \in \mathbb{R}^d$
**Ensure:** Shapley values $\phi_1^{\boldsymbol{x}}, \ldots, \phi_d^{\boldsymbol{x}}$
 1: Retrieve training data $X$, coefficients $\boldsymbol{\alpha}$, and kernel function $k$
 2: Compute feature-wise kernel vectors $\boldsymbol{z}_j = k_j(x_j, X_{:,j})$, for all $j = 1, \ldots, d$
 3: Precompute Shapley weights $\mu(q) = \frac{q!(d-q-1)!}{d!}$, for $q = 0, \ldots, d-1$
 4: **(Global scaling and characteristic polynomial)**
 5: $s \leftarrow \max_{i,j} z_{ij}$ {set $s = 1$ if all entries are zero}
 6: $\tilde{\boldsymbol{z}}_j \leftarrow \boldsymbol{z}_j/s$ for all $j = 1, \ldots, d$
 7: Initialize coefficient list Coeff with Coeff $= [\mathbf{1}]$
 8: **for** $j = 1$ **to** $d$ **do**
 9:    Let **new_coeff** be an empty list
10:    **new_coeff**$[0] \leftarrow -\tilde{\boldsymbol{z}}_j \odot$ Coeff$[0]$
11:    **for** $m = 1$ **to** len(Coeff) $- 1$ **do**
12:       **new_coeff**$[m] \leftarrow$ Coeff$[m-1] - \tilde{\boldsymbol{z}}_j \odot$ Coeff$[m]$
13:    **end for**
14:    **new_coeff**$[$len(Coeff)$] \leftarrow$ Coeff$[$len(Coeff) $- 1]$
15:    Coeff $\leftarrow$ **new_coeff**
16: **end for**
17: **(Synthetic division for leave-one-out ESPs)**
18: **for** $j = 1$ **to** $d$ **do**
19:    Let $r_j = \tilde{\boldsymbol{z}}_j$
20:    Initialize $Q^{(j)}[0..d-1]$
21:    $Q^{(j)}[d-1] \leftarrow$ Coeff$[d]$
22:    **for** $m = d - 1$ **to** $1$ **do**
23:       $Q^{(j)}[m-1] \leftarrow$ Coeff$[m] + r_j \odot Q^{(j)}[m]$
24:    **end for**
25:    $\{Q^{(j)}(t) = \prod_{i \neq j}(t - \tilde{\boldsymbol{z}}_i)\}$
26:    **Extract ESPs:**
27:    **for** $q = 0$ **to** $d - 1$ **do**
28:       $e_q(\mathcal{Z}_{-j}) \leftarrow (-1)^q Q^{(j)}[d-1-q] \cdot s^q$
29:    **end for**
30:    $\psi_j \leftarrow \sum_{q=0}^{d-1} \mu(q) \odot e_q(\mathcal{Z}_{-j})$
31:    $\phi_j^{\boldsymbol{x}} \leftarrow \boldsymbol{\alpha}^\top ((\boldsymbol{z}_j - \mathbf{1}) \odot \psi_j)$
32: **end for**
33: **return** $(\phi_1^{\boldsymbol{x}}, \ldots, \phi_d^{\boldsymbol{x}})$

---

**Proof**  Using the efficiency axioms for $\hat{v}_{\boldsymbol{x}}$, the results will follow.

## F.2 EXPLANATION ADDITIVITY FOR MMD

**Lemma 9.** *For the MMD with product kernel, the sum of Shapley values satisfies:*

$$\sum_{j=1}^d \phi_j^{MMD} = \widehat{MMD}^2(\mathbb{P}, \mathbb{Q}).$$

**Proof**  By the efficiency property of Shapley values (Shapley, 1953):

$$\sum_{j=1}^d \phi_j^{\text{MMD}} = v_{\text{MMD}}(\mathcal{D}) - v_{\text{MMD}}(\emptyset)$$

For the product-kernel decomposition:

$$v_{\text{MMD}}(\mathcal{D}) = \frac{1}{n(n-1)} \sum_{i \neq j} k(\boldsymbol{x}^{(i)}, \boldsymbol{x}^{(j)}) + \frac{1}{m(m-1)} \sum_{i \neq j} k(\boldsymbol{y}^{(i)}, \boldsymbol{y}^{(j)}) - \frac{2}{nm} \sum_{i,j} k(\boldsymbol{x}^{(i)}, \boldsymbol{y}^{(j)}) = \widehat{\text{MMD}}^2(\mathbb{P}, \mathbb{Q})$$

For the empty coalition (all features removed):

$$v_{\text{MMD}}(\emptyset) = \frac{1}{n(n-1)} \sum_{i \neq j} 1 + \frac{1}{m(m-1)} \sum_{i \neq j} 1 - \frac{2}{nm} \sum_{i,j} 1 = \frac{n(n-1)}{n(n-1)} + \frac{m(m-1)}{m(m-1)} - \frac{2nm}{nm} = 0$$

Thus, $\sum_{j=1}^{d} \phi_j^{\text{MMD}} = \widehat{\text{MMD}}^2(\mathbb{P}, \mathbb{Q}) - 0 = \widehat{\text{MMD}}^2(\mathbb{P}, \mathbb{Q})$. $\qquad\square$

### F.3 EXPLANATION ADDITIVITY FOR HSIC

**Lemma 10.** *For the HSIC dependence measure with a product kernel, the sum of Shapley values satisfies:*

$$\sum_{j=1}^{d} \phi_j^{HSIC} = \widehat{HSIC}(X, y).$$

**Proof** By the efficiency property of Shapley values (Shapley, 1953), the sum of Shapley values equals the difference between the value of the grand coalition and the empty coalition:

$$\sum_{j=1}^{d} \phi_j^{\text{HSIC}} = v_{\text{HSIC}}(\mathcal{D}) - v_{\text{HSIC}}(\emptyset).$$

From Definition 6, we have:

$$v_{\text{HSIC}}(\mathcal{D}) = \frac{1}{(n-1)^2} \text{tr} \left( \mathbf{HLH} \bigodot_{j \in \mathcal{D}} \mathbf{K}_j \right) = \widehat{\text{HSIC}}(X, y),$$

$$v_{\text{HSIC}}(\emptyset) = \frac{1}{(n-1)^2} \text{tr}(\mathbf{HLH11}^\top).$$

One can simply realize that $\mathbf{HLH11}^\top = 0$ as the sum of rows and columns in $H$ is zero, and substituting these into the efficiency property completes the proof. $\qquad\square$

## G HSIC ATTRIBUTION WITH TWO MULTIVARIATE VARIABLES

We studied the Shapley value computation for HSIC when it measures dependence between a multivariate variable $\boldsymbol{x}$ and a univariate target $y$. We now extend this framework to two multivariate variables $X \in \mathbb{R}^d$ and $Z \in \mathbb{R}^d$ with product kernels function $k$ and $l$. Given a sample $\{(\boldsymbol{x}^{(i)}, \boldsymbol{z}^{(i)})\}_{i=1}^{n} \sim \mathbb{P}(X, Z)$, HSIC$(X, Z)$ can be estimated as:

$$\widehat{\text{HSIC}}(X, Z) = \frac{1}{(n-1)^2} \text{tr} \left( \mathbf{HKHL} \right),$$

where $\mathbf{K} \in \mathbb{R}^{n \times n}$ is the kernel matrix computed using the kernel $k$, i.e, $\mathbf{K}_{ij} = k(\boldsymbol{x}^{(i)}, \boldsymbol{x}^{(j)})$, $\mathbf{L} \in \mathbb{R}^{n \times n}$ is the kernel matrix computed using the kernel $l$, i.e, $\mathbf{L}_{ij} = l(\boldsymbol{z}^{(i)}, \boldsymbol{z}^{(j)})$, and $\mathbf{H} = I - \frac{1}{n}\mathbf{11}^\top$ is the centering matrix ensuring zero mean in the feature space.

To address the attribution to variables in both $X$ and $Z$, we establish two cooperative games to attribute dependence contributions: For product kernels $k$ and $l$, we define two value functions for the two games:

(i) We first assume that $\mathbf{L}$ is fixed, and try to attribute the total HSIC to the variables in $X$. Since $k$ is a product kernel, we can define a value function similar to Definition 6. Let $\mathcal{D}_X$ be the set of variables of $X$, we therefore define the value function for attributing the total HSIC to $X$ variables as:

$$v_{\text{HSIC}_X}(\mathcal{S}) = \frac{1}{(n-1)^2} \text{tr} \left( \mathbf{HK}_{\mathcal{S}} \mathbf{HL} \right), \quad \forall \mathcal{S} \subseteq \mathcal{D}_X. \tag{7}$$

(ii) By the same token, we take $\mathbf{K}$ fixed and try to attribute the total HSIC to the variables in $Z$. Since $l$ is a product kernel, we define the value function as:

$$v_{\text{HSIC}_Z}(\mathcal{S}) = \frac{1}{(n-1)^2} \text{tr}\left(\mathbf{HKHL}_{\mathcal{S}}\right), \quad \forall \mathcal{S} \subseteq \mathcal{D}_Z, \tag{8}$$

where $\mathcal{D}_Z$ is the set of variables of $Z$.

Building on the two value functions, we can compute Shapley values for variables in $X$ and $Z$ separately. We denote the Shapley value of variable $j$ for $X$ and $Z$ as $\phi_j^{\text{HSIC}_X}$ and $\phi_j^{\text{HSIC}_Z}$, respectively. These values are interpreted as:

- $\phi_j^{\text{HSIC}_X}$ quantifies the contribution of the $j^{th}$ variable in $X$ to the total dependence between $X$ and $Z$;
- $\phi_j^{\text{HSIC}_Z}$ quantifies the contribution of the $j^{th}$ variable in $Z$ to the total dependence between $X$ and $Z$.

## H EXPERIMENTS

### H.1 EXPERIMENTAL SETUP

**SVM Optimization Using Optuna (Akiba et al., 2019)**   When using SVM in our experiments, we optimized the Support Vector Machine (SVM) with a Radial Basis Function (RBF) kernel using Optuna (Akiba et al., 2019), a robust hyperparameter optimization framework. The target type (either 'regression' or 'classification') was determined to guide the selection of the appropriate SVM model (`SVR` for regression and `SVC` for classification). The hyperparameters `C` and `gamma`, critical for the RBF kernel's performance, were optimized within an extensive range using a log-uniform distribution. Specifically, we defined the hyperparameters: `C` between $10^{-5}$ and $10^5$ and `gamma` between $10^{-5}$ and $10^3$, and utilized 5-fold cross-validation to ensure reliable evaluation. The optimization process aimed to minimize the mean squared error for regression tasks and maximize accuracy for classification tasks. After conducting the specified number of trials (n=100), the best hyperparameters were used to train a final SVM model on the entire dataset, yielding both the optimal model configuration and the best cross-validation score achieved during the optimization process.

**Training Gaussian Process (GP) Using K-Fold Cross-Validation**   When using GP, we trained a model using k-fold cross-validation to ensure robust evaluation and generalization performance. We defined the GP kernel as `C(1.0, (1e-4, 1e1)) * RBF(1.0, (1e-4, 10))`, suitable for both regression and classification tasks. For classification problems, `GaussianProcessClassifier` was utilized, while `GaussianProcessRegressor` was used for regression tasks. We employed `K-Fold` with $K = 5$ for cross-validation to evaluate the model's performance across different folds. All the hyperparameters, including the kernel width of the RBF kernel, are determined in the training process using optimization. During the cross-validation process, the model was trained on each fold, and predictions were made on the validation fold. Performance metrics were chosen based on the problem type: accuracy for classification models and mean absolute percentage error (MAPE) for regression models. The scores from each fold were aggregated to compute the average and standard deviation of the scores.

### H.2 EXECUTION TIME

To assess the computational efficiency of our recursive algorithm, we conducted a simulation study using a randomly generated kernel function of the form $\boldsymbol{\alpha}^\top k(\boldsymbol{x}, \mathbf{X})$ with 1000 samples. We computed Shapley values under both the brute-force enumeration and our recursive method by varying the number of features, as plotted in Figure 1. With a runtime budget of 300 seconds, the brute-force computation was feasible only for up to 30 features, already requiring more than 200 seconds at this scale, while exceeding the budget beyond that. In contrast, our recursive algorithm consistently completed the same computations in a fraction of the time, from milliseconds at lower dimensions to under 100 seconds, even with 500 features. These results clearly highlight the substantial efficiency gains and scalability of our method for high-dimensional settings.

## H.3 MMD EXPERIMENTS

In addition to the synthetic experiments for MMD, we first provide another experiment for the cases when there is no distribution discrepancy. To that end, $X$ and $Z$ are sampled from the same multivariate normal distribution across all 20 variables. The MMD is near zero, indicating the distributions are equivalent. Shapley values are computed and replicated 1000 times, with histograms plotted for each variable in Figure 6. The near-identical distributions of Shapley values across all variables reflect the uniform contribution of these variables to the MMD close to zero, consistent with the absence of any significant difference between the distributions.

We extend our analysis to the UCI Diabetes dataset, consisting of 442 samples and 10 baseline variables, including age, sex, body mass index (BMI), average blood pressure, and six blood serum measurements (shown by s1 to s6 features). The dataset is split into male and female subsets using the second variable (sex), which is excluded from the analysis, leaving nine variables for comparison.

Using MMD, we calculate the dissimilarity between male and female groups and then compute Shapley values to attribute variable contributions to the MMD. Figure 7 displays the Shapley values for the nine variables in the Diabetes dataset. The results show that $s3$ and $s4$ contribute most significantly to the MMD, followed by $bp$, $s6$, $age$, $s5$, and $s2$. In contrast, $bmi$ and $s1$ reduce the MMD, indicating their alignment across the two groups.

To validate these results, we analyze the marginal distributions of variables for males and females, as shown in Figure 8. The analysis confirms that variables with distinct marginal distributions between males and females (e.g., $s3$ and $s4$) have high positive Shapley values, reflecting their role in increasing the MMD. Conversely, variables with similar distributions (e.g., $s1$) exhibit negative Shapley values, highlighting their role in reducing the MMD.

## I HOW DOES PKEX-SHAPLEY DIFFER FROM MOHAMMADI ET AL. (2025B)?

In Mohammadi et al. (2025b), the authors propose an exact Shapley computation method for stochastic attribution in FANOVA GP models. Similar to PKeX-Shapley, their approach leverages Newton's identities and symmetric polynomial representations to obtain closed-form attributions. However, the two methods differ in both scope and capability.

PKeX-Shapley is designed specifically for kernel methods with product kernels. It presents a new value function, i.e., a functional baseline value function, to compute Shapley values. While this formulation enables exact computation of the mean Shapley values, it does not extend naturally to higher-order moments. In particular, when applied to Gaussian processes, computing the variance or covariance structure of Shapley values is nontrivial, since orthogonality does not generally hold for product kernels. As a result, our method cannot be directly extended to explain GP models in a stochastic way.

By contrast, Mohammadi et al. (2025b) focus on FANOVA GPs, whose kernels admit a functional ANOVA decomposition. Compared to product kernels, this class of kernels is more restrictive and often harder to train in practice. Nonetheless, the additional structure provides a key advantage: it enables exact polynomial-time computation not only of the mean but also of the variance and covariance of stochastic Shapley values—something that product kernels cannot achieve. This distinction highlights the central difference between their method and ours.

## J HOW DOES PKEX-SHAPLEY DIFFER FROM RKHS-SHAP (CHAU ET AL., 2022)?

RKHS-SHAP is the first kernel method-specific SHAP-based algorithm. While the author still employs the conditional expectation value function $\tilde{\nu}_{\boldsymbol{x}}(S) = \mathbb{E}[f(X) \mid X_S = \boldsymbol{x}_S]$, they used the fact that for function $f$ in the RKHS $\mathcal{H}_k$, it is possible to estimate $\tilde{\nu}_{\boldsymbol{x}}(S)$ non-parametrically utilizing a tool known as conditional kernel mean embedding. Specifically, this leads to the following expression:

$$\tilde{\nu}_{\boldsymbol{x}}(\mathcal{S}) = \boldsymbol{\alpha}^\top \mathbf{K}(\mathbf{K}_\mathcal{S} + n\lambda I)^{-1} k_\mathcal{S}(\mathbf{X}_\mathcal{S}, \boldsymbol{x}_\mathcal{S})$$

which of course, is different from our functional baseline value function

$$\nu_{\boldsymbol{x}}(\mathcal{S}) = \boldsymbol{\alpha}^\top k_\mathcal{S}(\mathbf{X}_\mathcal{S}, \boldsymbol{x}_\mathcal{S}).$$

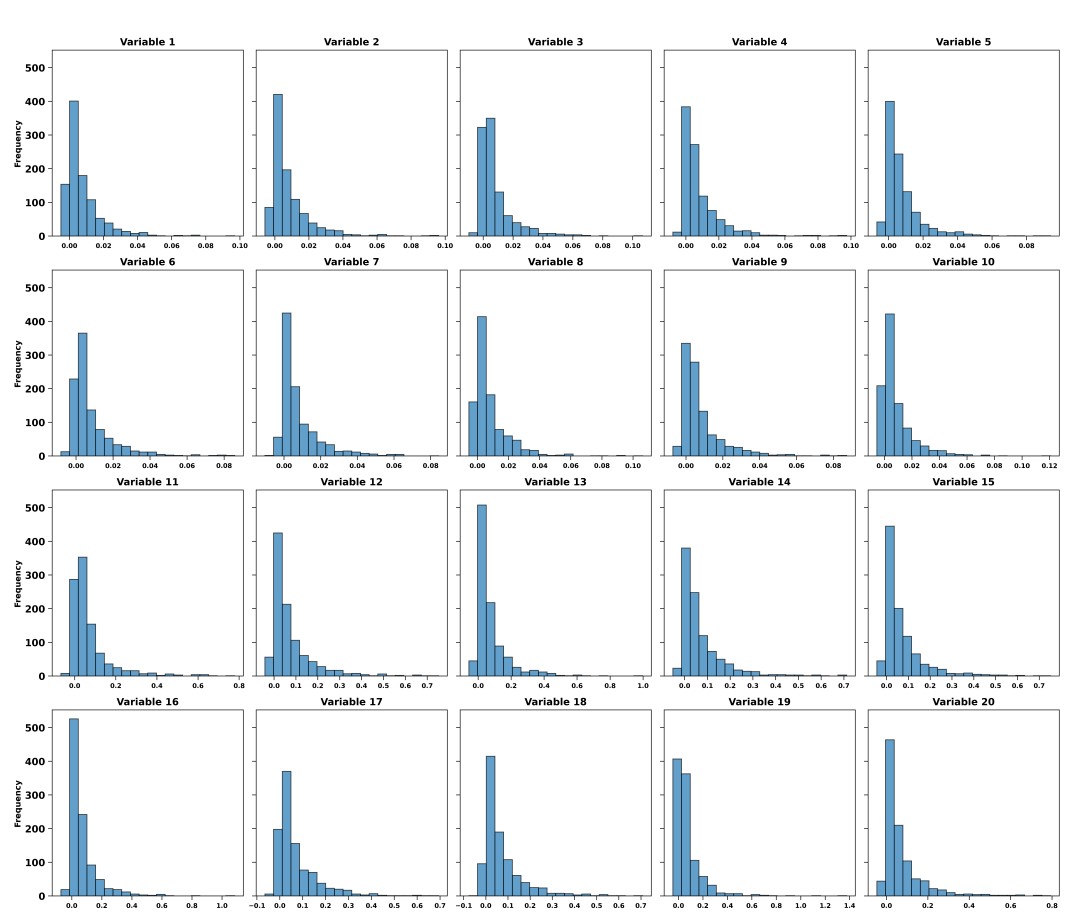

Figure 6: Shapley values for the synthetic data sets with equal distributions. All variables contribute equally to the near-zero MMD.

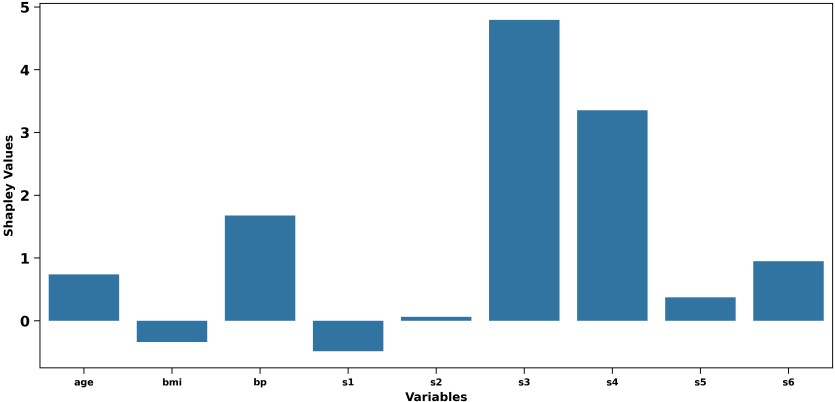

Figure 7: Shapley values explaining MMD between male and female subsets in the UCI Diabetes data set.

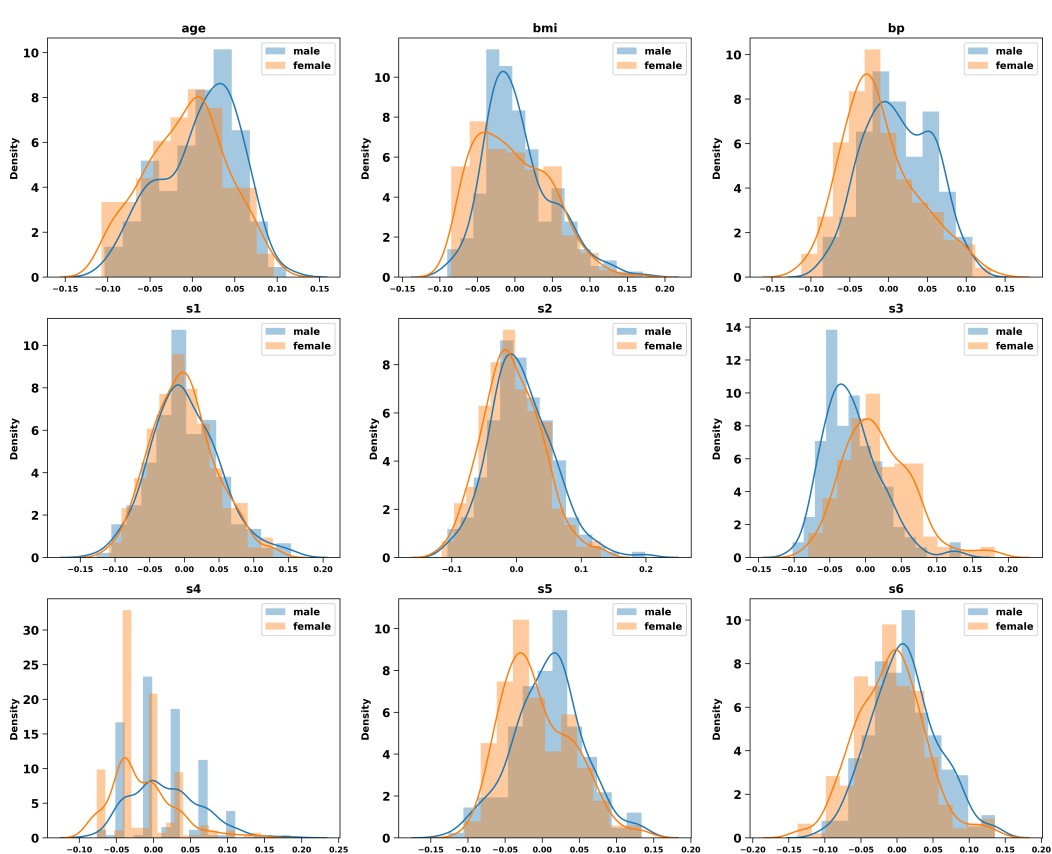

Figure 8: Marginal distributions of variables for male and female subsets in the UCI Diabetes dataset.

Although looking different in their expression, there is an interesting mathematical connection that is not apparent at first glance. First of all, we need to understand that conditional expectations can be interpreted as orthogonal projections to the space of nice-behaving functions defined on the conditioning variable. Specifically, let $\mathcal{L}^2(\mathcal{X})$ be the space of square-integrable functions on $X$, then for a random variable $Y$, the conditional expectation $\mathbb{E}[Y \mid X]$ is the unique element in $\mathcal{L}^2(\mathcal{X})$ that is closest to $Y$ in mean-square error:

$$\mathbb{E}[Y \mid X] = \arg \min_{Z \in \mathcal{L}^2(\mathcal{X})} \mathbb{E}[(Y - Z)^2].$$

As a result, this interpretation of conditional expectations, and thus of the conditional-expectation-based value function $\tilde{\nu}_{\boldsymbol{x}}$, allows us to directly connect it to our functional baseline value function. As established in Proposition **??** and Appendix **??**, the latter can also be viewed as an orthogonal projection, albeit onto a different function space.

