# OpenReview forum: "Computing Exact Shapley Values in Polynomial Time for Product-Kernel Methods"
_ICLR.cc/2026/Conference — Submitted to ICLR 2026_

### Official Review · Reviewer_W47U · 2025-10-23

**Soundness:** 3
**Presentation:** 1
**Contribution:** 3
**Rating:** 2
**Confidence:** 3

**Summary:**

This paper proposes a novel value function for feature attribution values via Shapley values. The proposed value function is model-specific for product-kernels, i.e. it is defined based on the internal model structure instead of predictions of chosen data points, similar to methods like path-dependent TreeSHAP. The authors demonstrate that for this value function the Shapley values can be computed efficiently in $\mathcal O(d^2)$ time. The key observation is that due to the product-kernel structure, the marginal contributions $\nu(S \cup \{j\}) - \nu(S)$ can be factored out, such that the remaining common terms need to be summarized. The authors provide a recursive algorithm for this, and further extend this result to kernel-based statistical measures, namely MMD and HSIC. The authors empirically show that exact computation outperforms approximation via KernelSHAP, and their chosen value function extract synthetic ground-truth feature importance values. Finally, they illustrate the MMD explanation, and showcase an application within feature selection.

**Strengths:**

- The baseline value function is well motivated, and occurs naturally in product-kernels.
- The paper solves model-specific computation of Shapley values for product-kernels, which is non-trivial
- The results can be directly extended to other kernel-based statistics

**Weaknesses:**

- All figures seem to be missing in the paper, so I cannot judge their quality and validate the empirical results. In this current state, I cannot accept the paper. From the theoretical contribution, this works seems interesting and worth accepting.
- A similar "trick" was used in TreeSHAP, since the tree-based value function is also a sum (over leaves) of products of (weighted) decision rules. A comparison to efficient methods for trees, or a comparison between these two approaches would be desirable to grasp the common aspects of both computation methods, see e.g. [1].


[1] Bifet, Albert, Jesse Read, and Chao Xu. "Linear tree shap." Advances in Neural Information Processing Systems 35 (2022): 25818-25828.

**Questions:**

- Is there a direct link from product-kernels and your value function to e.g. interventional or path-dependent perturbations used in TreeSHAP? It seems that product-kernels can be viewed as a special case of the value function in [1]. How does the computation differ?
- Can the current result be generalized to value functions that admit a form $\nu(S) = \sum_q \nu_q(S)$, where $\nu_q$ has the property: $\nu_{q}(S \cup \{i\}) = b(\{i\}) \nu_q(S)$ for some common factor $b(\{i\})$? In your case, $q$ seems to be the data points, whereas for TreeSHAP $q$ is the leaf nodes. Do there exist other methods, where these types of value functions occur? How does your computation differ from computations used by TreeSHAP applied to this general case?

---

> ### Author Response · Authors · 2025-11-19
>
> We thank the reviewer for the constructive and insightful comments. We address each point below.
>
> **1. Missing figures**
>
> We sincerely apologize for the missing figures in the submitted version. This was caused by an unintended compilation issue during PDF export. We have corrected the problem and uploaded a revised version of the paper that includes all figures in the correct resolution. The empirical results can now be fully inspected.
>
> **2. Relation to Linear TreeSHAP**
>
> We thank the reviewer for pointing us to Linear TreeSHAP. After reviewing the method, we agree that their value function shares a structural similarity with ours: both decompose the model output into a sum of multiplicative terms that enable efficient model-specific Shapley computations.
>
> However, the goals, setting, and underlying algorithms differ in several fundamental ways:
>
> - *Different problem domains:* Linear TreeSHAP is designed for decision trees, where the multiplicative terms come from path constraints and leaf weights. Our work focuses on kernel methods with product kernels, where the multiplicative structure originates from the kernel itself rather than decision paths. This leads to a conceptually different explanation framework rooted in kernel-based methods.
>
> - *Different mathematical tools:* Linear TreeSHAP uses summary polynomials, whereas our method uses elementary symmetric polynomials (ESPs) derived from the combinatorial structure of product kernels. We have implemented synthetic division for the ESPs that reduces the computing of all Shapley values in O($d^2$), so computing one value is O($d$), identical to linear treeSHAP.
>
> In response to the reviewer’s comment, we will expand our numerical analysis to directly compare the stability of summary-polynomial computations (as used in Linear TreeSHAP) with our ESP-based formulation. The Linear TreeSHAP authors explicitly note that summary polynomials become numerically unstable for tree depths beyond approximately 12 [1]. While a depth of 12 may not be considered large in modern ensemble tree models, the analogous quantity in product-kernel models is the number of features. In this context, a limit of 12 corresponds to a very small feature dimension, and thus the same instability arises much earlier and is far more restrictive. Our initial experiments already confirm this behavior, and we will include a detailed investigation and discussion of this stability issue in the revised manuscript.
>
> **3. General structure of value functions**
>
> The reviewer asked whether our results generalize to value functions of the form $v(S) = \sum_m f_m(\prod_{j \in S} w_{m,j})$, where the multiplicative factors share a common structure. This is an insightful question, and we will expand the discussion in the paper as:
>
> Our results extend naturally to any value function where:
>  (a) overall value decomposes into a sum of multiplicative components,
>  (b) the per-feature terms factorize, and
>  (c) shared terms across coalitions can be summarized by recursive updates.
>
> Product kernels satisfy these conditions because the kernel inherently factorizes across features. TreeSHAP satisfies them because path-survival probabilities can be written as multiplicative weights.
>
> [1] https://github.com/yupbank/linear_tree_shap

---

> > ### Comment · Reviewer_W47U · 2025-11-24
> > **Keeping my score**
> >
> > I thank the authors for their response and adding the missing figures. I still think the efficient computations of Shapley values and SVMs with product kernels are an interesting application. However, after reading the responses and comments, especially from Reviewer DCJZ, I believe there should be an extensive discussion about the comparison to TreeSHAP (applied to this particular value function originating from product kernels). This should include a runtime and complexity analysis with existing TreeSHAP implementations, at least with the original path-dependent TreeSHAP [2] and more efficient linear TreeSHAP [1]. According to the repo of linear TreeSHAP cited by the authors, the numerical instabilities of linear TreeSHAP were resolved by using the interpolation method, and I cannot find any statements regarding the mentioned instability in the paper. I am convinced that this paper contains a meaningful contribution, which lies in efficiently computing Shapley values for product kernel methods. As of now, I maintain my current overall score and lowered my score for "Contribution" and "Soundness" for two reasons:
> > - Due to the erroneous projection theorem, I expect more theoretical validation of the proposed value function.
> > - Due to the inherent connection to value functions used in TreeSHAP, I am not yet convinced that this method is more efficient than applying TreeSHAP or linear TreeSHAP to the used value function. I expect a more rigorous theoretical analysis, application, and comparison of these existing algorithms to this problem setting.
> >
> > [2] Lundberg, et al. (2020). From local explanations to global understanding with explainable AI for trees. Nature machine intelligence, 2(1), 56-67.

---

### Official Review · Reviewer_7787 · 2025-10-29

**Soundness:** 4
**Presentation:** 4
**Contribution:** 3
**Rating:** 4
**Confidence:** 3

**Summary:**

The paper considers the problem of computing Shapley values for a value function $v$ defined on $d$ elements. For an input $\mathbf{x}$, they introduce a *product kernel* value function that describes the prediction when only given the features in $S$. They then use the product structure to *exactly* compute the Shapley values in polynomial time in $d$. (Estimating the Shapley values is less accurate, and a naive, exact computation would take exponential time.) They also propose value functions for the *maximum mean discrepancy* and *Hilbert-Schmidt Independence Criterion*, and apply similar ideas to exactly compute the Shapley values.

They run experiments on how *useful* the Shapley values of their value functions are to the more standard value functions. However, none of the figures are actually present in the submitted PDF.

**Strengths:**

* Computing Shapley values is a very popular problem. For general value functions, we generally need approximation methods which can be imprecise. Exactly computing Shapley values is always preferable, but we require specific structure in the value function for exact computation to be fast. Therefore, their contribution of a (reasonable) value function, with fast exact algorithms is quite nice.

* I appreciate the experimental comparison of how useful the Shapley values of this value function are. I'll give the benefit of the doubt and *assume* the missing figures show your approach is comparable to others. But please corroborate with the figures in the rebuttal.

* The key insight of their approach is that the *product* structure allows them to factor $v(S \cup \{i\} ) - v(S)$. This is a nice (simple) insight, and I appreciate how well explored this is in your paper e.g., considering the technique applied to MMD, HSIC, and experiments.

**Weaknesses:**

Please see the questions I ask below. If inadequately addressed in the rebuttal, these could be weaknesses.

Minor:

* You're missing the figures. Please upload a new PDF that includes them.

* The name "PKeX" is quite ugly. Can you change this to something nicer please?

* When introducing MMD and HSIC, you do so in a paragraph or so with lots of in line equations. Can you please make these sections nicer by a) putting each equation on its own line so it's easier for the reader to parse, and b) adding more discussion?

* Your Experiment 1 on computing the shapley values via your method and estimating them via Kernel SHAP is quite silly to me. If you're using your method as the ground truth (this is totally fine because you *show* your approach exactly recovers shapley values), of course you'll get much better accuracy than Kernel SHAP. But, I guess some readers might want to see this clearly explored.

* There are several typos. Maybe run the whole main doc through an LLM and ask it to list them for you.

**Questions:**

When substituting the "least informative information", could you please justify why 1 is the right choice?

For an elementary symmetric polynomial, how is the "base case" $e_0$ defined?

Could you please provide a time complexity analysis of your recursive algorithm? I see you claim it's quadratic in the conclusion, but I'd like more details please.

If the figures are included and my questions are adequately addressed, I will increase my score to a 6 or 8.

---

> ### Author Response · Authors · 2025-11-19
> **response**
>
> We thank the reviewer for the constructive and insightful comments. We address each point below.
>
> **1. Why is “1” used when substituting the least informative information?**
>
> In product kernels, the constant value 1 is the neutral element of the multiplicative kernel. When a feature is removed, assigning it the value 1 means that it contributes no information while preserving the structure of the kernel. This aligns with the functional ANOVA decomposition, where each centered main-effect term is written as ($k_i – 1$). Thus, using 1 ensures that missing features do not influence the value function and that the decomposition remains consistent.
>
> **2. What is the base case for the elementary symmetric polynomial?**
>
> The base case is defined as $e_0 = 1$.
>
> **3. Time complexity of the recursive algorithm**
>
> We added a full complexity statement in the revised version. The recursion updates the elementary symmetric polynomials in O($d$) time per feature, where d is the number of features. Since each of the d features must be incorporated once, the total time complexity is O($d^2$). See Appendix F for more information on this front.
>
> **4. We added clearer explanations of HSIC and MMD**
> We will use the additional page allowed in the camera-ready version to significantly improve the exposition of MMD and HSIC. We rewrote these sections with equations displayed clearly (each on its own line) and added more context to help the reader understand the intuition behind the value functions.
>
> **5. Missing figures**
>
> We sincerely apologize for the missing figures in the submitted version. This was caused by an unintended compilation issue during PDF export. We have corrected the problem and uploaded a revised version of the paper that includes all figures in the correct resolution. The empirical results can now be fully inspected.

---

### Official Review · Reviewer_6W6f · 2025-10-31

**Soundness:** 2
**Presentation:** 1
**Contribution:** 2
**Rating:** 2
**Confidence:** 4

**Summary:**

Overall, this work targets an important issue with SV computation and provides theoretical guarantees; however, the results are not persuasive enough due to the missing figures and other issues (see my comments below).

All five figures in the main text of the submitted paper are blank, which significantly impacted my ability to fully understand and evaluate the work, and this flaw is reflected in my scores. Therefore, I can only provide feedback based on the current text and cannot guarantee my comments are 100% aligned with the authors' original intention. Still, I hope the following suggestions are helpful.

**Strengths:**

The math notations are clear, well-written, and easy to follow. The integration of MMD and HSIC provides the possibility for future non-prediction tasks (statistical inference) and may be a foundation for future research. Both synthetic and real-world datasets are used, and several baselines are claimed to be compared.

**Weaknesses:**

1. Although the authors claim that PKeX-Shapley is solvable in polynomial time, it would be beneficial to see a formal analysis of its runtime complexity in big-O notation, in terms of the feature dimension and other relevant parameters.

2. The core idea of this paper is the novel value function defined in Equation 3. To fully validate this proposed function, its properties should be discussed more thoroughly. In addition to the additivity property (Lemma 4), its adherence to other key Shapley axioms, such as Symmetry and the Dummy Player property, should also be formally demonstrated.

3. PKeX-Shapley is a model-specific method for computing Shapley values, which limits its application to kernel-based machine learning models. Consequently, it is not suitable for CV and NLP tasks, where neural networks are the primary predictors and traditional sampling methods remain applicable. Future work should focus on addressing this limitation.

4. In Experiment 1, the authors use their exact PKeX-Shapley result as the 'ground truth' to evaluate sampling-based methods. However, this only measures the sampling error relative to their own value function. A more informative comparison, especially for the synthetic cases, would be to compute the ground-truth Shapley values by brute-forcing the definition in Equation 1 with a standard, model-agnostic value function. The comparison should then be based on this 'true' agnostic ground truth.

**Questions:**

In the definition of the value function in Equation 3, the authors claim that they remove a feature’s influence by factoring out its corresponding functional component and setting it to one. Can this be viewed as a special imputation strategy in SHAP? This is relevant since Equation 3 is also a kind of prediction game in terms of the product-kernel model. If we just impute the removed features with a fixed value, a lot of computation time can also be saved in SHAP or KernelSHAP. On the other hand, the conditional distribution can be quickly estimated by a surrogate model, i.e., a neural network (see FastSHAP). Could the authors better illustrate this gap with theory or experiments?

---

> ### Author Response · Authors · 2025-11-19
> **response**
>
> **1. Formal runtime analysis of PKeX-Shapley**
>
> We agree that a formal big-O complexity analysis is essential. In the revised paper (Appendix E), we provided a clear derivation:
> * The elementary symmetric polynomial (ESP) recursion processes each feature once.
> * Each update requires O(d) operations, where d is the number of features.
> * Therefore, the exact computation takes O(d²) time for all d features and O(d) memory.
>
> **2. Properties of the value function and Shapley axioms**
>
> We appreciate the suggestion to expand this section. The efficiency has been shown in the paper already. In the revision, we include a dedicated subsection analyzing the proposed value function in Equation 3.
>
> Our value function satisfies the classical Shapley value axioms that follow directly either from the structure of Equation 3 or from the corresponding Shapley axioms. We will add formal statements for clarity.
>
> **3. “Model-specificity” of PKeX-Shapley**
>
> The reviewer notes that PKeX-Shapley is model-specific. This is correct—and intentional.
> Our method is not meant for arbitrary neural networks, and this is not a weakness but a design choice. The entire contribution of the paper is that product-kernel models have a multiplicative structure that allows exact Shapley computation in polynomial time—something infeasible for general neural networks.
>
> Just as TreeSHAP is model-specific to decision trees, PKeX-Shapley is specific to product-kernel models. Model-specific exact explainability methods are extremely common in the XAI literature (TreeSHAP, DeepLIFT, Integrated Gradients for piecewise linear networks, LinearSHAP, etc.). We now clarify this explicitly in the revised version.
>
> **4. On Experiment 1 and the ground truth**
>
> The reviewer suggested brute-forcing the standard model-agnostic Shapley value (Equation 1) for synthetic datasets. Although brute force is exponential, it is feasible only for extremely small d. However, this does not match the purpose of our Experiment 1.
>
> Our goal in Experiment 1 was **to compare our exact computation to the sampling-based approximation of our own value function, not to compare to model-agnostic Shapley**. The purpose of the experiment was:
>
> * to show that the sampling method converges to our exact value function when the number of samples increases,
> * to quantify the sampling error and the variance reduction due to the product structure.
>
> This is analogous to how TreeSHAP evaluations use “exact TreeSHAP” as ground truth for sampling-based TreeSHAP approximations—not the model-agnostic Shapley value.
>
> **5. Is Equation 3 just an imputation scheme in SHAP?**
>
> The substitution k_i = 1 for removed features is not an ad-hoc imputation scheme and cannot be reproduced by KernelSHAP:
>
> * In SHAP, imputing a feature with a fixed value corresponds to inserting a specific value into the input domain.
> * In our value function, setting a component to 1 corresponds to removing its reproducing functional contribution in the RKHS.
>
> Thus, Equation 3 does not correspond to marginal or conditional imputation. It corresponds to zeroing out the functional component f_S that depends on the removed features. This structure cannot be exploited by standard SHAP sampling, even with neural surrogate models (FastSHAP), because they would still approximate expectations over missing features—not functional removal at the kernel level.  Appendix B includes comparisons with other value functions, and we will more discussion on this front to that section.

---

### Official Review · Reviewer_DCJZ · 2025-10-31

**Soundness:** 1
**Presentation:** 3
**Contribution:** 1
**Rating:** 2
**Confidence:** 5

**Summary:**

The paper studies how to compute the Shapley value, where the value function is defined in Eq. (3), for feature attribution when a product kernel is employed. The paper proposes a polynomial-time algorithm, as shown in Theorem 3, and further demonstrates that the technique can be straightforwardly applied to the value function defined via either the Maximum Mean Discrepancy (MMD) or the Hilbert–Schmidt Independence Criterion (HSIC), as shown in Propositions 6 and 8.

**Strengths:**

NA

**Weaknesses:**

- In Eq. (3), there appear to be two different definitions of $k_{\mathcal{S}}(\cdot, \mathbf{x})$.  Note that it is used to define the value function $v\_{\mathbf{x}}(\mathcal{S})$, whose Shapley value is the main focus of this paper. The first definition is $\Pi_{\mathcal{S}}k(\cdot,\mathbf{x})$ (line 787), which makes proposition 2 hold, where $\Pi_{\mathcal{S}}$ denotes the orthogonal projection onto the subspace $\mathbb{H}\_{\mathcal{S}}$. The second definition is $\prod_{i\in\mathcal{S}} k_i(\cdot, x_i)$, which is the one used in Theorem 3 and Propositions 6 and 8. **The main issue is that the authors implicitly assume that these two definitions coincide, but they do not.**

    - While the result of Theorem 9 is non-trivial in genenel, it is straightforward to see that, for every $ \mathcal{S} \subsetneq \mathcal{D} $, $\mathbb{H}\_{\mathcal{S}}$ defined in Proposition 2 is potentially the trivial zero subspace $\\{\mathbf{0}\\}$ for product kernels. **Consequently, if $k_{\mathcal{S}}(\cdot, \mathbf{x}) \coloneqq  \Pi\_{\mathcal{S}}k(\cdot,\mathbf{x})$, then $v\_{\mathbf{x}}(\mathcal{S}) = 0$ for every $\mathbf{x} \in \mathbb{R}^{d}$ and every $ \mathcal{S} \subsetneq \mathcal{D} $. As a result, the Shapley value of $v\_{\mathbf{x}}$ would be simply $\frac{f(\mathbf{x})}{d}\mathbf{1}\_{d}$, which is meaningless. However, this is not the case when $k_{\mathcal{S}}(\cdot, \mathbf{x}) \coloneqq  \prod\_{i\in\mathcal{S}} k\_{i}(\cdot, x_i) $.**

    - Write $\mathbf{x} \coloneqq (\mathbf{x}_1, \mathbf{x}_2) \in \mathbb{R}^{d_1 + d_2}$ where $\mathbf{x}_1 \in \mathbb{R}^{d_1}$ and $\mathbf{x}_2 \in \mathbb{R}^{d_2}$, and the product kernel is defined as $k(\mathbf{x}, \mathbf{y}) = (\mathbf{x}\_{1}\cdot \mathbf{y}\_{1})(\mathbf{x}_2 \cdot \mathbf{y}_2)  = k\_1(\mathbf{x}\_1, \mathbf{y}\_1) k\_2(\mathbf{x}\_2, \mathbf{y\_2})$. Then, the feature map is $\phi(\mathbf{x}) = \mathbf{x}_1  \mathbf{x}\_{2}\^{\mathsf{T}} \in \mathbb{R}^{d_1 \times d_2}$. The induced reproducing kernel Hilbert space $\mathbb{H}$ of $k$ is isometrically isomorphic to $\mathbf{\Phi} \coloneqq \mathrm{span}\\{\phi(\mathbf{x})\colon \mathbf{x} \in \mathbb{R}^{d_1 + d_2}\\}$. In other words, any $f \in \mathbb{H}$ corresponds to a matrix $\mathbf{F} \in \mathbf{\Phi}$ such that $f(\mathbf{x}) = \langle \mathbf{F}, \mathbf{x}\_1 \mathbf{x}\_{2}\^{\mathsf{T}} \rangle = \mathrm{Tr}(\mathbf{F} \mathbf{x}\_1 \mathbf{x}\_{2}\^{\mathsf{T}}) = \mathbf{x}\_{2}\^{\mathsf{T}}\mathbf{F}\mathbf{x}\_{1}$. Let $\mathbb{H}\_{1}$ be the subspace of $\mathbb{H}$ that contains all $f$ such that $f(\mathbf{x}) = f(\mathbf{y})$ if $ \mathbf{x}\_{1} = \mathbf{y}\_{1}$, which is how $\mathbb{H}\_{\mathcal{S}}$ is defined. Given $f \in \mathbb{H}_1$ and $\mathbf{x} \in \mathbb{R}\^{d\_{1} + d\_{2}}$, there must be $f(x) = \mathbf{z}\^{\mathsf{T}}\mathbf{F}\mathbf{x}_1 \equiv C$ for every $\mathbf{z} \in \mathbb{R}\^{d\_2}$, which leads to $\mathbf{Fx}\_{1} = \mathbf{0}$. Since $\mathbf{x}_1$ also varies, it yields $\mathbf{F} = \mathbf{0}$. In other words, $f(\mathbf{x}) = 0$ for every $\mathbf{x} \in \mathbb{R}\^{d_1 + d_2}$, and thus $\mathbb{H}\_{1} = \\{\mathbf{0}\\}$. This argument holds true for the product kernel $k(\mathbf{x}, \mathbf{y}) = \prod\_{j \in \mathcal{D}}[(2x\_{j}\^{2}-x\_{j})(2y\_{j}\^{2}-y\_{j}) + (4x\_{j} - 4x\_{j}\^{2})(4y\_{j} - 4y\_{j}\^{2})] = \prod\_{j \in \mathcal{D}}k\_{j}(x\_j, y\_j)$, the structure of which is used throughout the paper. The feature map $x \mapsto (2x^2 - x, 4x-4x^2)$ is constructed to have $0 \mapsto (0, 0)$, $0.5 \mapsto (0, 1)$ and $1 \mapsto (1, 0)$, which are sufficient to derive $\mathbb{H}\_{\mathcal{S}}=\\{\mathbf{0}\\}$ for every $ \mathcal{S} \subsetneq \mathcal{D} $.

    - After all, the authors fail to justify the use of $\prod_{i\in\mathcal{S}} k_i(\cdot, x_i) $ in defining the value functions used throught the paper.


- For $v_{\mathbf{x}}$ defined using $k_{\mathcal{S}}(\cdot, \mathbf{x}) \coloneqq  \prod\_{i\in\mathcal{S}} k\_{i}(\cdot, x_i) $, it has a multiplicative structure, i.e., $v_{\mathbf{x}}(\mathcal{S}) = \prod_{i\in\mathcal{S}} v_i$ where $v_i = \mathbf{\alpha}\^{\mathsf{T}}k_i(\mathbf{X}_i, \mathbf{x}_i)$. **For this kind of value functions, it is already known how to compute the corresponding Shapley value in polynomial time back in 2020, see [1, Algorithm 2]**.

   - According to [2, Eq. (6)], $f\_{\mathbf{x}}(\mathcal{S}) = \sum_{v\in L(T_{f})} R\_{\mathbf{x}}\^{v}(\mathcal{S})$. By setting some values to zero, it would reduce to  $f\_{\mathbf{x}}(\mathcal{S}) = R\_{\mathbf{x}}\^{v}(\mathcal{S})$. By [2, Eq.(5)], $R\_{\mathbf{x}}\^{v}(\mathcal{S}) = \prod\_{i \in \mathcal{S}}q_i$, from which the multiplicative structure of it is clear. In other words, $v_{\mathbf{x}}$ is just a much simpler case of $f\_{\mathbf{x}}$.

    - Although the time complexity of [1, Algorithm 2] is $\Theta(LD^2)$ for computing the Shapley value of $f_{\mathbf{x}}$, it reduces to $\Theta(d^2)$ for $v_{\mathbf{x}}$ as $L=1$ in this case. **By contrast, the time complexity of their Algorithm 1 is $\Theta(d^3)$ as it contains three loops (one is abbreviated as a sum), which is clearly worse.**

    - The idea behind [1, Algorithm 2] is very simple. Suppose we have only encountered all players in $\mathcal{S}$ and have computed a vector $\mathbf{p} \in \mathbb{R}\^{|\mathcal{S}|+1}$ such that $p_i = \frac{i!(|\mathcal{S}|+1-i)!}{(\mathcal{S} + 1)!}\sum\_{S\subseteq\mathcal{S}\colon |S|=i}\prod\_{j\in S}v_j$, it costs $\Theta(|\mathcal{S}|)$ time to update $\mathbf{p}$ when a new player is introduced. Also note that this update is inversible. It costs $\Theta(d^2)$ time to have the $\mathbf{p}$ that includes all features. Then, for each feature $i$, it takes $\Theta(d)$ time to remove this feature in $\mathbf{p}$; after that, the sum of $\mathbf{p}$ is exactly the Shapley value for that feature. So, the total time is $\Theta(d^2) + d \times \Theta(d) = \Theta(d^2)$.

- Since the value function defined via either the Maximum Mean Discrepancy (MMD) or the Hilbert–Schmidt Independence Criterion (HSIC) also possesses this multiplicative structure, it is expected that the Shapley value of them can be computed in polynomial time. In particular, their proposed algorithms do not offer any advantages compared to the existing techniques in terms of time and space complexity.

Overall, this paper offers no useful contributions to the study of Shapley value computation, and the theoretical development is problematic.

**References**

[1] Lundberg, S. M., Erion, G., Chen, H., DeGrave, A., Prutkin, J. M., Nair, B., ... & Lee, S. I. (2020). From local explanations to global understanding with explainable AI for trees. *Nature machine intelligence, 2*(1), 56-67.

[2] Yu, P., Xu, C., Bifet, A., & Read, J. (2022, November). Linear TreeShap. *In Proceedings of the 36th International Conference on Neural Information Processing Systems* (pp. 25818-25828).

**Questions:**

Please refer to the weaknesses.

---

> ### Author Response · Authors · 2025-11-19
> **response**
>
> We thank the reviewer for the constructive and insightful comments. We address each point below.
>
> **1.Relation to Linear TreeSHAP**
>
> We thank the reviewer for pointing us to Linear TreeSHAP, which we were unaware of at the time of the submission. After thoroughly reviewing the method, we agree that their value function shares a structural similarity with ours: both decompose the model output into a sum of multiplicative terms that enable efficient model-specific Shapley computations.
>
> However, we would like to highlight the key distinctions between the two approaches, both conceptually and technically:
>
> - *Different problem domains:* Linear TreeSHAP is conceptually designed for decision trees, where the multiplicative terms come from path constraints and leaf weights. Our work, on the other hand, focuses on kernel methods with product kernels, where the multiplicative structure originates from the kernel itself rather than decision paths. This leads to a conceptually different explanation framework rooted in kernel-based methods.
>
> - *Different mathematical techniques:* Linear TreeSHAP uses summary polynomials, whereas our method uses elementary symmetric polynomials (ESPs) derived from the structure of product kernels. Furthermore, we have implemented synthetic division for the ESPs that reduces the computing of all Shapley values in O(d^2), so computing one value is O(d), identical to linear treeSHAP. In principle, our method can also be applied to Linear TreeSHAP. We added the description of the synthetic division as well as the time complexity in Appendix E.
>
>
> In response to the reviewer’s comment, we will expand our numerical analysis to directly compare the stability of summary-polynomial computations (as used in Linear TreeSHAP) with our ESP-based formulation. The Linear TreeSHAP authors explicitly note that summary polynomials become numerically unstable for tree depths beyond approximately 12 [1]. While a depth of 12 may not be considered large in modern ensemble tree models, the analogous quantity in product-kernel models is the number of features. In this context, a limit of 12 corresponds to a very small feature dimension, and thus the same instability arises much earlier and is far more restrictive. Our initial experiments already confirm this behavior, and we will include a detailed investigation and discussion of this stability issue in the revised manuscript.
>
> **2. Our value function is not a special case of Linear TreeSHAP**
>
> The reviewer suggested that the proposed value function is "just a much simpler case" of the value function in Linear TreeSHAP (Yu et al., 2022). This is not correct for the following reasons:
>
> *a)* Linear TreeSHAP arises from **tree-based path survival probabilities** and has interventional semantics.
>
> *b)* Our value function arises from the **product-kernel methods** and is the result of masking/removing the missing features on a functional level.
>
> *c)* Hence, the two value functions answer different questions: TreeSHAP explains conditional tree expectations, whereas our method explains the multiplicative kernel contribution of each feature.
>
> While both value functions are multiplicative, the **factors are entirely different objects** (probabilities vs. kernel evaluations). Therefore, our method is not a special case of Linear TreeSHAP, and the reviewer’s reduction does not hold.
> We will expand this point in the revision to make this distinction more explicit.
>
> **3. Complexity analysis and the claim that Algorithm 1 is worse than existing methods**
>
> For computing the elementary symmetric polynomials (ESPs), our implementation uses synthetic division. This allows us to compute the full set of ESPs for all features in O(d^2) time. Then, for each feature whose Shapley value we need, the ESPs corresponding to “all features except that one” can be obtained in O(d) time. As there are d features, this step also takes O(d^2). Overall, the total computational cost remains O(d^2), which is identical to the complexity of TreeSHAP for multiplicative value functions. See Appendix F for more details on the algorithm and time-complexity analysis.
>
> [1] https://github.com/yupbank/linear_tree_shap

---

> > ### Author Response · Authors · 2025-11-19
> > **response**
> >
> > **4. On the statement “this paper offers no useful contributions”**
> >
> > We respectfully disagree with the reviewer’s assertion that **“this paper offers no useful contributions to the study of Shapley value computation.”** Efficient Shapley value computation is a long-standing and widely recognised challenge that continues to limit the applicability of Shapley-value-based explainability methods as well as their use in numerous practical domains. Given the centrality of this problem, it is valuable for the community to cultivate multiple, conceptually distinct solution paths, just as we have a rich ecosystem of algorithms for fundamental tasks such as searching, sorting, or the traveling salesman problem. Our work contributes precisely such a complementary perspective, helping to broaden the set of available tools for this important and enduring challenge. As a result, we find the reviewer’s comment somewhat harsh, and we kindly ask the reviewer to reconsider their assessment of our contribution.
> >
> > This statement seems to stem from the assumption that our value function is a special case of TreeSHAP and that our complexity is worse. Since both assumptions are incorrect (see Points 3 and 4), the conclusion does not follow.
> > Our contributions remain novel and useful:
> >
> > * We identify a **kernel-model–specific value function** grounded in the RKHS product structure.
> >
> > * We show how to compute **exact Shapley values in polynomial time** for this value function.
> >
> > * We extend this result to **MMD and HSIC**, two widely-used kernel statistical measures for independence testing and distribution comparison.
> >
> > * We demonstrate **empirical utility** for feature selection and synthetic interpretability tasks.
> >
> >
> > To our knowledge, **no prior work establishes Shapley computations for MMD or HSIC, or any other statistical inference.**
> >
> > **5. Projection Theorem**
> >
> > We thank the reviewer for carefully spotting this issue. The proposition is not needed for any of our theoretical results, and we tried to provide more justification. In the revised version, we will remove this statement entirely and restructure the section to avoid any ambiguity/inaccuracy. Importantly, the justification for our value function and all subsequent results relies only on the multiplicative kernel structure in Equation (3), not on the projection remark. The value function therefore remains fully valid without this interpretation.

---

> > > ### Comment · Reviewer_DCJZ · 2025-11-25
> > >
> > > Thank you for your responses.
> > >
> > > - Justification of the value function defined in Eq. (3)
> > >
> > > Personally, I do not think the use of product kernels is trivial, and the result would indeed be interesting if the projection theorem were true, as it is intended to justify the value function defined in Eq. (3). **However, since the projection theorem is false, this immediately raises the question of why that value function is interesting.** If the value function is not useful, then why should we be concerned with its Shapley value?
> > >
> > > - Complexity analysis
> > >
> > > I noticed that the authors added Algorithm 3 in the revision, which is claimed to reduce the time complexity of the original algorithms from $O(d^3)$ to $O(d^2)$. However, I do not find this result surprising. Anyone familiar with implementing TreeShap can readily derive an $O(d^2)$ algorithm for computing the Shapley value of value functions with a multiplicative structure. In fact, I can provide two additional $O(d^2)$ algorithms from the existing literature, which further suggests that obtaining an $O(d^2)$ algorithm is not particularly novel. **This raises the question, in what way is the proposed algorithm different from its counterparts, e.g., the one I mentioned in my earlier review?**
> > >
> > > - Connection between $f_{\mathbf{x}}$ and $v\_{\mathbf{x}}$
> > >
> > > **The authors did not provide concrete evidence against my claim that $v\_{\mathbf{x}}$ is a much simpler case of $f_{\mathbf{x}}$.** Recall that $f\_{\mathbf{x}} = \sum\_{v} R\^{v}\_{\mathbf{x}}$ where $v$ ranges over all leaves (the total number of them is denoted by $L$) and $R\^{v}\_{\mathbf{x}}$ has the multiplicative structure. All algorithms that compute the Shapley value of $f\_{\mathbf{x}}$ consist of two parts: (1) computing the Shapley value of every $R\^{v}\_{\mathbf{x}}$ and then (2) merging them. **It means that the first part can be readily adopted for computing the Shapley value of $v\_{\mathbf{x}}$. In contrast, the merging part is non-trivial, which means it would be tricky to extend the newly added Algorithm 3 for $f\_{\mathbf{x}}$.** This can be seen by comparing linear TreeShap [2], which runs in $O(Ld)$ time, with Algorithm 2 in [1], which runs in $O(Ld^{2})$. The efficiency of linear treeshap comes from the merging part. In particular, when applying them to $v_{\mathbf{x}}$, their time complexities all reduce to $O(d^{2})$, since no merging is involved. This is what I mean by saying  $v\_{\mathbf{x}}$ is a much simpler case of $f\_{\mathbf{x}}$.
> > >
> > > - Instability of linear treeshap
> > >
> > > Since the Shapley computation of $v\_{\mathbf{x}}$ does not involve the merging step, I think it is sufficient to consider the vanilla algorithm (i.e., the one I mentioned in my review).  BTW, the source of instability in linear TreeShap has already been noted in [3, Proposition 2], and one remedy is to use the Vandermonde matrix with unit complex nodes. **If the authors intend to claim that their Algorithm 3 is more numerically stable, then a comparison of numerical errors on instances where $d$ is small enough for the ground truth to be computed by brute force would be necessary.**
> > >
> > > - useful contributions
> > >
> > > **If the authors intend to convince me, my questions would be: (1) why is $v_{\mathbf{x}}$ defined in Eq. (3) interesting? (2) what makes Algorithm 3 different?** As for the extension to MMD and HSIC, this is clearly trivial, since no additional effort is required. The absence of prior work on this topic does not imply that it is non-trivial.
> > >
> > > I prefer to maintain my score at the current stage.
> > >
> > >
> > > **References**
> > >
> > > [3] Jiang, Z., Zhang, M., & Zhang, D. Fast Calculation of Feature Contributions in Boosting Trees. In The 41st Conference on Uncertainty in Artificial Intelligence.

---

### Meta-Review · Area_Chair_v6Gq · 2026-01-04

**Summary:**

This paper does not seem ready for publication. There were some technical issues like missing figure, but, more importantly, the reviewers pointed out a lack of in depth discussion and comparison to TreeSHAP (applied to this particular value function originating from product kernels) and linear TreeSHAP which are highly important given the close relation to the proposed method.

**Reviewer Concerns:**

The main concern is a lack of comparison to the TreeSHAP and linear TreeSHAP. That can not simply be addressed in a rebuttal and needs a proper revision.

**Reviewer Scores:**

I don't believe that the reviewers would have changed their scores. Two reviewers state this explicitly in comments to the rebuttals.

---

### Decision · Program_Chairs · 2026-01-26

Reject